

# Delineation of marine ecosystem zones in the northern Arabian Sea using an objective method

Saleem Shalin[1,*], Annette Samuelsen[2], Anton Korosov[2], Nandini Menon[1], Björn C. Backeberg[2,3,4], and Lasse H. Pettersson[2]

[1]Nansen Environmental Research Centre (India), Kochi, India
[2]Nansen Environmental and Remote Sensing Center, Bergen, Norway
[3]Coastal Systems Research Group, Natural Resources and the Environment, Council for Scientific and Industrial Research, Stellenbosch, South Africa
[4]Nansen-Tutu Centre for Marine Environmental Research, Department of Oceanography, University of Cape Town, South Africa

*Correspondence to:* Shalin S. (shalinsaleem@gmail.com)

**Abstract.** The spatial and temporal variability of marine autotrophic abundance, expressed as chlorophyll concentration, is monitored from space and used to delineate the surface signature of marine ecosystem zones with distinct optical characteristics. An objective zoning method is presented and applied to satellite-derived Chlorophyll a (Chl-a) data from the northern Arabian Sea (50°–75° E and 15°–30° N) during the winter months (November – March). Principal Component Analysis (PCA)

and Cluster Analysis (CA) were used to statistically delineate the Chl-a into zones with similar surface distribution patterns and temporal variability. The PCA identifies principal components of variability and the CA splits these into zones based on similar characteristics. Based on the temporal variability of Chl-a pattern within the study area, the statistical clustering revealed six distinct ecological zones. The obtained zones are related to the Longhurst provinces to evaluate how these compared to established ecological provinces. The Chl-a variability within each zone was then compared with the variability of oceanic

and atmospheric properties viz. mixed-layer depth (MLD), wind speed, sea-surface temperature (SST), Photosynthetically Active Radiation (PAR), nitrate and Dust Optical Thickness (DOT) as an indication of atmospheric input of iron to the ocean. The analysis showed that in all zones, peak values of Chl-a coincided with low SST and deep MLD. Rate of decrease in SST and deepening of MLD are observed to trigger the intensity of the algae bloom events in the first four zones. Lagged cross-correlation analysis shows that peak Chl-a follows peak MLD and SST minima. The MLD time-lag is shorter than the SST lag

by eight days, indicating that the cool surface conditions might have enhanced mixing, leading to increased primary production in the study area.

An analysis of monthly climatological nitrate values showed increased concentrations associated with the deepening of the mixed-layer. The input of iron seems to be important in both the open ocean and coastal areas of the northern and northwestern part of the Northern Arabian Sea, where the seasonal variability of the Chl-a pattern closely follows the variability of

iron deposition.





## 1 Introduction

The Northern Arabian Sea is a dynamic ocean area, where upwelling, downwelling, convective overturning, mesoscale eddies, fronts and planetary waves commonly occur. The ocean dynamics are significantly influenced by the seasonal monsoon cycles

(Rao et al., 2010; Schott and McCreary, 2001). Seasonality in marine primary production in the Arabian Sea associated with the monsoon was studied by Lévy et al. (2007), who showed that two distinct seasonal bloom patterns occur: one during winter and another during summer. During the winter monsoon period, convective overturning is common in the area enhancing nutrient supply to the ocean surface and increasing biological productivity (Madhupratap et al., 1996). Iron is found to be a limiting nutrient and is primarily supplied through atmospheric fallout of desert dust in this region (Banerjee and Kumar, 2014; Johansen

et al., 2003; Moffett et al., 2015; Naqvi et al., 2010; Wiggert and Murtugudde, 2007). Under cloud-free conditions optical sensors onboard satellites measure spectral reflectance of ocean surface from which chlorophyll a (Chl-a) concentration can be derived, which serves as a proxy for phytoplankton biomass (Pettersson and Pozdnyakov, 2013; Wiggert et al., 2002). However, the accuracy of Chl-a retrieval is low in turbid waters and regions where the satellite signal is hampered by unaccounted atmospheric influences (Pozdnyakov and Grassl, 2003; Kahru et al., 2014). In this work, which focuses on open-ocean waters

away from turbid coastal waters, we anticipate that such detrimental factors are not important.

Classification of the ocean into ecological zones is a useful tool to understand the interactions between physical and biochemical marine processes as well as the interactions between the surrounding water masses and zones.Longhurst (1995, 1998, 2006) described the global ocean in terms of several ecological provinces, considering the entire plankton ecology in relation to regional meteorological and oceanographic conditions. A similar approach by Spalding et al. (2012) classified global pelagic

waters into 37 large-scale pelagic provinces based on oceanographic properties. Both the Longhurst and the Spalding provinces are static representations of the global ocean based on an annual cycle. Devred et al. (2007) proposed a method of classification that allows for seasonal movements of boundaries of the ecological provinces. They used satellite measurements of Chl-a and Sea-Surface Temperature (SST) from different seasons to re-define dynamic provinces in the northwest Atlantic Ocean. Dynamic variations in global biogeochemistry based on Chl-a, surface salinity and temperature were examined by Reygondeau et

al. (2013) who observed that seasonal as well as inter-annual variability influenced the delineation of the provinces.

In this study Chl-a satellite remote sensing data from the winter seasons (November to March) were used to delineate the marine ecological zones to study phytoplankton variability and its drivers in the northern Arabian Sea. Though we know that significant primary production occurs in summer in the Arabian Sea, it is also very cloudy that there are insufficient remote sensing observations to perform the analysis. The winter season was chosen as it represents the period when, due to cloud-free

conditions, high quality satellite data are available and high values of Chl-a ($> 0.5$ mg.m$^{-3}$) prevailed in the study area. Apart from this temporal restriction, as the proposed method utilises satellite data, only surface coverage information is available. However, a significant larger spatio-temporal quantity of data is available for the delineation study, compared to usage of in situ observations. The fact that during the study period (winter), deep chlorophyll maxima is weak and shallow due to light



attenuation by surface chlorophyll (Mignot et al., 2014) allows the usage of surface Chl-a alone as a proxy for columnar Chl-a content.

In each identified zone, Chl-a is averaged for each winter month for the study period in order to understand its variability. Similarly, the time series of zonal averages of environmental factors viz. SST, Mixed-Layer Depth (MLD), Photosynthetically Active Radiation (PAR) and wind speed are calculated and compared with Chl-a to understand their influence on marine primary productivity. To this end, we also examine time-lagged correlation of Chl-a with SST and MLD. The influence of nitrate and dust optical thickness (DOT) on phytoplankton variability is also analysed.

## 2   Data

This study utilizes satellite-derived data on surface Chl-a concentration, Photosynthetically Available Radiation (PAR), Sea Surface Temperature (SST) and Aerosol Optical Thickness for derivation of Dust Optical Thickness (DOT). These quantities derived by remote-sensing are supplemented with other environmental properties, including surface winds from reanalysis, modelled MLD, and climatological monthly nitrate concentrations, as described in detail below.

### 2.1   Chlorophyll-a data (Chl-a)

Global gridded Chl-a concentrations at 9 km spatial resolution, based on the MODIS Aqua sensor are available from NASA's ocean colour data portal (http://oceandata.sci.gsfc.nasa.gov). The present work uses monthly, climatological and 8-day composite Chl-a data from November to March during the winter seasons from 2002 to 2013. The MODIS Chl-a algorithm derives the near-surface Chl-a concentration (expressed in mg m$^{-3}$), from remote-sensing reflectance (Werdell and Bailey, 2005). The climatological dataset is used for the zoning procedure. Monthly data are used in the time series analysis and time-lagged correlations are computed using 8-day composites.

### 2.2   Photosynthetically Available Radiation (PAR)

PAR is the quantum energy flux from the sun in the visible spectrum (expressed in Einstein m$^{-2}$ day$^{-1}$). Under cloud-free conditions PAR is calculated from radiance measurements at the top of the atmosphere derived from satellite remote sensing data in the visible spectral range and corrected for the effects of clouds (Frouin et al., 1995). PAR used in this study is also available from the above-mentioned ocean-colour data portal of NASA.

### 2.3   Sea Surface Temperature (SST)

We used MODIS Aqua day time, 8-day, composite SST at a spatial resolution of 9 km, available from NASA's ocean colour data portal. The SST is derived from radiance signals in the thermal infra-red at 11 µm and 12 µm, from the satellite sensor. The brightness temperatures are derived from the observed radiances by inversion (in linear space) of the radiance versus blackbody temperature relationship (Haines et al., 2007).



## 2.4 Dust Optical Thickness (DOT)

DOT used is calculated utilizing the method of Kaufman et al. (2005) and is given as:

$$DOT = \frac{AOT(f_{an} - f) - AOT_{ma}(f_{an} - f_{ma})}{(f_{an} - f_{du})} \tag{1}$$

where, AOT is the Aerosol Optical Depth, which is obtained from MODIS / Aqua (http://oceancolor.gsfc.nasa.gov). AOD

represents total aerosol content in the atmospheric column, while DOT indicates just the dust content in the atmospheric

column. Here, 'f' is the fraction of AOT contributed by fine particles. Suman et al. (2014) reported 'f' to be 0.25 over the

northern Indian Ocean. The quantities $f_{an}$, $f_{ma}$ and $f_{du}$ are respectively the fine-mode fractions of anthropogenic aerosol,

maritime aerosols and dust. Following the work of Nair et al. (2005) and Banerjee and Kumar (2014), $f_{an}$ is taken as 0.90.

Similarly, $f_{ma}$ is assumed to be 0.47, $f_{du}$ is set at 0.25. The fma value is an average value for the period of 2003–2011 over the

western part of the Equatorial Indian Ocean and $f_{du}$ is based on satellite values during dust outbreaks in the Middle East. Also,

$AOT_{ma}$ is the maritime AOT, calculated according to (Smirnov et al., 2003), as,

$$AOT_{ma} = 0.007 * w + 0.05 \tag{2}$$

where $w$ is the wind speed in m s$^{-1}$. This study used wind at 10 metres obtained from ERA-Interim reanalysis.

## 2.5 Winds

The ERA-Interim reanalysis data of 12-hourly wind components at 10 m simulated by atmospheric model from the European

Centre for Medium-Range Weather Forecasts (ECMWF) at 1.0°x1.0° spatial resolution was retrieved (Dee et al., 2011). These

ERA-Interim wind fields were used to calculate the wind speed.

## 2.6 Mixed Layer Depth (MLD)

The MLD for the Northern Arabian Sea used in this work is defined as the depth where the temperature is 1°C colder than

that at the surface temperature (Kumar and Narvekar, 2005). In this study the vertical temperature-profile data are weekly

averages from a Hybrid Coordinate Ocean Model (HYCOM) simulation for the Indian Ocean (Bleck, 2002; George et al.,

2010). HYCOM combines the optimal features of isopycnic-coordinate and fixed vertical grid ocean circulation models in

one framework. The adaptive (hybrid) vertical grid conveniently resolves regions of vertical density gradients, such as the

thermocline and surface fronts. A detailed analysis and validation of this ocean model for the Indian Ocean can be found in

(George et al., 2010).

## 2.7 Nitrate

Present study utilizes monthly climatological nitrate profiles available from NOAA National Centers for Environmental In-

formation (NCEI) / World Ocean Atlas 2013 (WOA 2013) (http://www.nodc.noaa.gov). WOA 2013 includes global nutrient

profiles at one-degree spatial resolution, which is the average of all unflagged interpolated values from all available in-situ ob-

servations (Garcia et al., 2013). Climatological data of nitrate used in this study are objectively analysed values for each winter



month, such that nitrate availability in each zone is calculated by averaging nitrate values within the mixed layer determined from HYCOM model.

## 3 Method for delineation of ecological zones

A method to delineate the study area objectively into ecological zones as per statistically distinct surface Chl-a characteristics
is developed. The method is based on the sequential application of Principal Component Analysis (PCA) and Cluster Analysis (CA) to series of satellite-derived images of surface Chl-a concentration.

### 3.1 Principal Component Analysis (PCA)

PCA is a statistical method that uses orthogonal transformation to identify the principal components (PCs) contributing to the variance of a signal. This method normalizes the dataset and computes covariances, eigenvectors and corresponding eigenvalues
for each PC. The eigenvectors are then sorted by decreasing eigenvalues (Abdi and Williams, 2010). The first PC is oriented in the direction of the largest variation of the original variables and passes through the centre of the data distribution. The second largest PC lies in the direction of the next largest variation, and passes through the centre of the data and is orthogonal to the first PC, and so forth.

### 3.2 Cluster analysis (CA)

K-means clustering is a signal processing method used to partition a given set of observation vectors into 'k' number of clusters, where k can be any integer greater than one (Kanungo et al., 2002). This method generates a set of centroids, one for each of the k clusters. Observation vectors are classified into clusters such that each observation vector is assigned to that cluster for which the total distance from vector to cluster centroid is minimum. For example, if a vector A is closer to centroid i than any other centroids, then A belongs to the cluster i.

## 4 Objective delineation of ecosystem zones in the northern Arabian Sea

Based on a monthly climatology (averaged over the years 2002-2013) of Chl-a concentration in the northern Arabian Sea for the winter months from November through March, five principal components of variability were obtained. The components account for respectively 80%, 13%, 7%, 4%, 2% and 1% of the variance in the monthly Chl-a distribution pattern.

Following the method of Z-score scaling (Johnson and Wichern, 1992) the values of principal components (PCs) were
scaled to one standard deviation centered on the mean. In addition, values of the first PC were converted so that the probability distribution of the values is closer to the Gaussian distribution (Figure 1) using the following equation (3) and (4).

$$P_{GAUSS} = log_{10}(min(P_0) - (P_0)) \hfill (3)$$

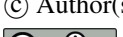



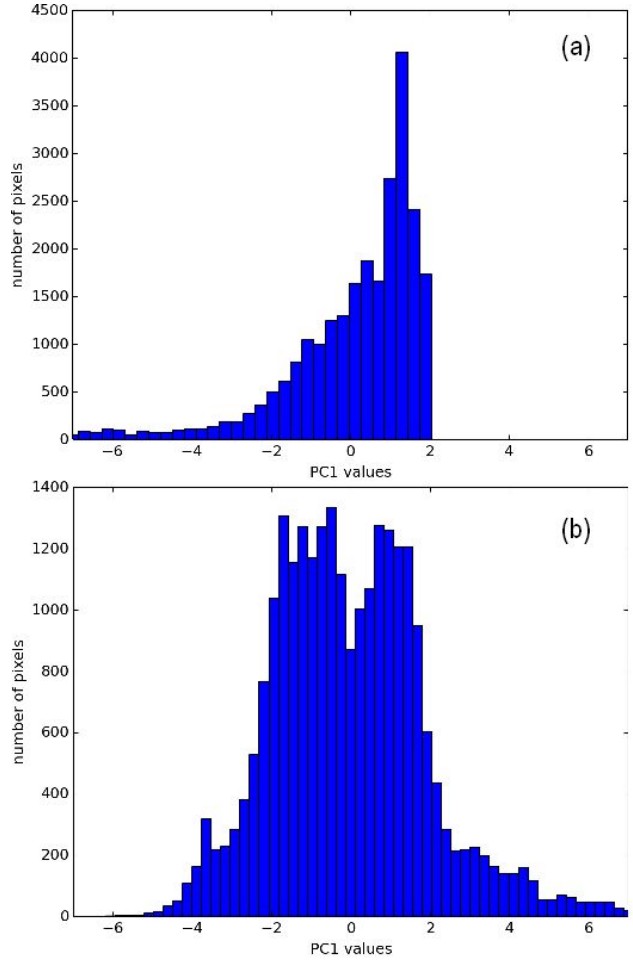

**Figure 1.** Probability distribution of PC1 before (a) and after the scaling and centering around the mean conversion (b).

$$P_{NORM} = \frac{(mean(P_{GAUSS}) - P_{GAUSS})\sigma_0}{\sigma_{GAUSS}} \qquad (4)$$

where $P_0$ denotes original value of the first principal component, $P_{GAUSS}$ denotes value of PC1 after conversion to a Gaussian distribution, $P_{NORM}$ denotes values of PC1 after scaling and centering around the mean, $\sigma_0$ denotes standard deviation of $P_0$ values and $\sigma_{GAUSS}$ denotes standard deviation of $P_{GAUSS}$ values.

Maps of principal components (PC1-5) are examined with regard to spatial distribution, information content and noise contamination (Figure 2). Ranges of $P_{NORM}$ decay from 8 (PC1), to 4 (PC2), to 2 (PC3), to 1 (PC4) and to 0.4 (PC5) confirming that most of information about spatial and temporal dynamics of Chl-a is retained in PC1. High values associated with PC1 are observed in the southern open ocean part of the study area whereas low values are observed along coastal areas of western India and near the coast of Oman. This indicates the difference between ecosystem dynamics in the oligotrophic waters (southern





**Figure 2.** Individual maps of principal components (PC 1 to 5) and RGB composite of the first three statistically significant components.

open ocean) and those in the coastal eutrophic waters (coastal and northern area). Such a northwestern and southeastern gradient has been observed by Prakash and Ramesh (2007) in the study area using satellite Chl-a. Jaswal et al. (2012) have reported a north – south gradient in the study area during winter, based on SST observations. Map of values associated with PC2 (Figure 2) shows that high values appear in the Gulf of Oman (mostly oligotrophic, but with significant seasonal variations) and the

5    shelf areas of Oman, Pakistan and western India (mostly mesotrophic, with low seasonal variations). The seasonal variation in the algal bloom pattern off Oman is attributed to iron availability (Wiggert and Murtugudde, 2007; Naqvi et al., 2010). The





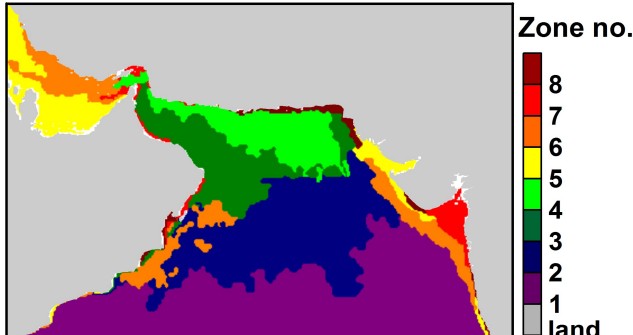

**Figure 3.** The identified ecological zones obtained from the combination of 4 PCs and 8 CAs.

spatial pattern of PC3 suggests that waters in southern part of Gulf of Oman (oligotrophic) differ from the mesotrophic waters near the coasts of Pakistan and Gujarat in agreement with Naqvi et al. (2010). The map of PC4 and PC5 shows a very low range of values and very high spatial inhomogeneity meaning that it contains mostly noise.

Visual inspection of RGB composite (Figure 2f) where values of PC1, PC2 and PC3 are displayed respectively using shades
of red, green and blue colours helps to identify zones with similar statistical attributes. Zones with a similar colors have similar combinations of PC values and therefore also similar seasonal dynamics of Chl-a. Such patterns allow the application of a semi-automated statistical clustering method to delineate the study region into areas with distinct Chl-a dynamics based on the values of principal components as discussed in section 3.

Several possible zoning maps were produced by varying the number of PCs and clusters in order to objectively delineate
the northern Arabian Sea into ecological zones (Appendix A). The final delineation into ecological zones was obtained by combining the first 4 PCs and 8 clusters (Figure 3), based on general Chl-a pattern in the Northern Arabian Sea. Spatial smoothing was applied to the selected zone map. The methodology used in zone map selection and smoothing procedure are provided in Appendix A. Satellite-derived Chl-a values along coastal and shallow waters are found to be erroneous, hence the coastal shallow water regions under zones 5, 7 and 8 as well as the part of zone 6 inside the Persian Gulf and patches along
Yemen coast are excluded from further analysis in this study. This leaves the first four zones and the region in zone 6 along Oman and the west coast of India. Zone 6 has two regions that lie on opposite sides of the Arabian Sea and the physical forcing affecting Chl-a concentration along the two regions is likely to be different. Therefore, these two regions are considered as separate ecological zones. As a result, a total of six distinct ecological zones are delineated in the study area (Figure 4a).

## 4.1 Comparison of ecological zones with Longhurst provinces

The six ecological zones identified fall into the two Longhurst provinces. These are respectively the northwest Arabian up-welling province (ARAB), covering the west and central part of the study area and western India coastal province (INDW) to the east of the study area. The border between these two Longhurst provinces is demarcated with a pink line in Figure 4a.





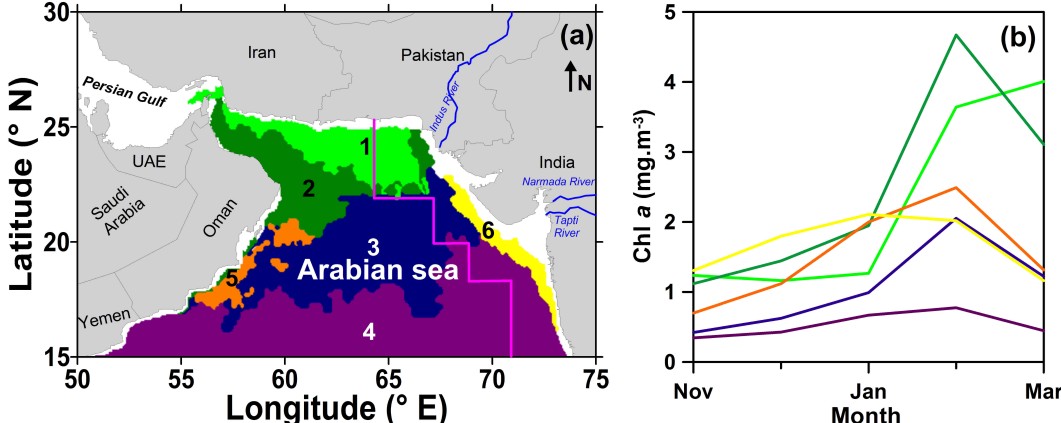

**Figure 4.** Map of the delineated ecological zones including the two Longhurst provinces in the Northern Arabian Sea. Pink line demarcates the border between the Longhurst provinces in the study area, respectively the northwest Arabian upwelling province to the west and the western India coastal province to the east. (b) The mean monthly climatology of surface Chl-a concentration during the winter months, for each zone, plotted using the same colours as in Figure 4(a) to represent each zone.

Zone 1 appears in the northern part of the study area (Figure 4a). During winter, moderate (5-10 m s$^{-1}$) northeasterly winds blows over the area. Intense cooling is reported in this region, which enhances primary production (Madhupratap et al., 1996; Kumar and Prasad, 1996). Similar to zone 1, intense cooling and high production also occur in zones 2 (Madhupratap et al., 1996). First three zones have similar Chl-a pattern, such that peak values occur either during the month of February or March

and for January Chl-a concentration is always half than its peak value. These three zones are strong upwelling regions with high values of Chl-a concentration during the winter (Banse and English, 2000). The cool and dry northeasterly winds blowing onto the region from the adjacent land territories enhance cooling at the ocean surface. Consequently, increased evaporation leads to a decrease in surface temperature and increase in surface salinity and density, creating convective mixing (Kumar and Prasad, 1996; Prasad and Ikeda, 2002; Shetye et al., 1992). All these three ecological zones are stretched across both the Longhurst

provinces, with the majority of the area located in the ARAB province. It is to be mentioned here that the ARAB province with upwelling in the Arabian Sea according to Longhurst's classification represent provinces with strong upwelling during summer and with strong convective cooling during winter. The southern part of the study area includes zone 4, where winter cooling is less intense and less marine production occurs, compared with zones 1-3 (Jyothibabu et al., 2010). However, similar to zones 1-3, zone 4 also is split across the two Longhurst provinces, with the western and central parts of zone 4 falling into the northwest

Arabian upwelling province and the eastern part of the zone into the west Indian coastal province. Zone 5 includes the coastal area along the Oman coast between 18°N-22.5°N and the coastal region along the west coast of India from 16°N to 23°N is included under zone 6. These coastal areas are highly productive during winter. The physical mechanisms in the northern and northwestern part of the Arabian Sea are very different from those in the eastern part. Strong convective mixing prevails in the northern and northwestern parts of the study area. The strong stratification in the east, due to the presence of low salinity





and high temperature water limits convective mixing in the eastern part of the study area (Naqvi et al., 2006). The eastern part comprises the zone 6, a part of zone 3 and zone 4. Among these three zones, zone 6 is a coastal area and is vulnerable to coastal complex processes. Nutrient supply from Narmada and Tapi rivers as well as atmospheric deposition of nitrogen enhances marine production in zone 6. For comparing Chl-a in six zones with Longhurst's province, we have classified zone 1,

zone 2, zone 3, zone 4 and zone 5 in the Longhurst ARAB province and zone 6 as the INDW province (Figure 4a). Maximum Chl-a observed during February is consistent in both provinces of Longhurst as well as the present six zones. During winter, ARAB (0.5-0.8 mg.m$^{-3}$) and INDW (0.4-0.6 mg.m$^{-3}$) have low values of Chl-a with similar range of variability (Longhurst, 2006). However, our study has identified high values of Chl-a concentration (>0.5 mg.m$^{-3}$) with significant gradients between various parts of the study area. From our analysis, it is clear that the northern parts has higher concentrations of Chl-a, which

decreasing concentrations towards south. Also the variation between each zone is identified higher concentrations of Chl-a in the western parts compared to the more easterly located zones. With only two Longhurst provinces such a spatial difference is not detectable. This spatial difference is due to the difference in physical mechanisms as mentioned in the above paragraph. Longhurst classification is based on 1° resolution global Chl-a maps in the context of regional meteorological and physical oceanographic variability (Longhurst, 1995). It also uses Chl-a observations from different time periods. In contrast, the present

study utilises primarily Chl-a concentration obtained from satellite sensors at about 100 times the resolution used by Longhurst for regional mapping and classification of ecological zones in the northern Arabian Sea. Hence, this regional classification could delineate the spatial Chl-a variability better and the obtained zones contain more detailed regional information. This study is restricted to the analysis of data for the winter season. This work intends to characterise a more complete delineation of ecological zones and the mechanisms driving marine production in the study area during winter. Therefore, the influence

of other ecological factors such as SST, MLD, PAR, wind and nutrients, on Chl-a production is included in the interpretations of the Chl-a pattern in each of the ecological zones. The Longhurst classification accounts for the differences in physical conditions between the northwestern and eastern parts. However, according to (Naqvi et al., 2006), downwelling in the eastern part in winter cannot extend to the northern boundary of West Indian coastal province of Longhurst. Since the present zonal classification limits zone 6 from extending to the entire northern portion of the Longhurst province, our zones seem to be

realistic in capturing regional Chl-a variability during winter season.

## 5   Time series analysis

Based on the magnitude of Chl-a in each zone, time series data of Chl-a and other environmental parameters (wind speed, MLD, PAR and SST) are examined to understand better the relations between physical and biological processes within each zone. Note that, the influence of water temperature on primary productivity through control of metabolism and respiration is

a highly non-linear process (Wetzel, 2001) and cannot be accounted for in the present study. Monthly climatology of Chl-a concentrations in the identified ecological zones have all moderate to high values (0.3-5.0 mg m$^{-3}$). Also, Chl-a follows a semi-cyclic seasonal variation pattern during the winter months with maximum values in February (Figure 4b). In zone 6, peak Chl-a is observed during January. Variability in the northern, most productive, part (zones 1 and 2) is discussed first and then





the southern, least productive, zones (zones 3 and 4) are considered. Finally, the time series along coastal and continental shelf zones including zones 5 and 6 are examined. The mean and standard deviation for each of these parameters are calculated for each winter month (November to March).

## 5.1 The ecological zones in the northern and most productive part of the Arabian Sea

In general, Chl-a concentration in zones 1 and 2 follows a typical wintertime cyclic variability with its peak values during the month of February (Figure 5c). Throughout the study period, the Chl-a concentration during February is at least double the concentration during January in these two zones. SST follows an inverse pattern compared with that of Chl-a, such that SST minima coincide with Chl-a maxima. Surface waters are relatively warm (>27°C) during November and cool as winter progress, with stronger cooling in zone 1 and 2 (Figure 5b). By January, SST has reduced by 2.5-3.0°C in both zones with a minimum of

23-24°C occurring in February. In March, the SST increases to 23-26°C. Although the Chl-a range is approximately the same for both zones, comparatively SST is lower in zone 1 than zone 2. The inverse relationship between SST and Chl-a have weak correlation coefficient [1] in zone 1 (r = 0.39, n=60) and zone 2 (r = 0.55, n=60).

    A deepening of the MLD during winter is seen in both zones (Figure 5b). During November, the MLD is shallow ( 35 m), and as winter progresses, MLD deepens to  80 m in January and in zone 1 to 90 – 110 m during February. In general, MLD in

zone 2 is 10 m shallower than in zone 1 during January and February. The MLD starts to shallow again in March. The peak concentrations of Chl-a coincide with the deepest MLD. However, MLD and Chl-a in zone 1 and 2 are moderately correlated (correlation coefficient, r = 0.28).

    During winter in the study area SST cooling initiates MLD deepening. Decrease in SST is mainly due to evaporation, which has dual effects i.e., increase in salinity and reduction in temperature, causing increased density of surface water (Naqvi et

al. 2006), as a consequence of which convective overturning takes place. As winter progresses, SST drops and convective overturning occurs (salinity and temperature effect), increasing the MLD (Shankar et al., 2015). MLD also influences SST variability. For example, when the MLD deepens, SST will decrease as cool water is mixed toward the surface. On the contrary, during a shallow MLD, SST is generally higher (Cronin and Kessler, 2002). Hence, both of these environmental parameters are dependent on each other and both influence marine primary production in the study area. Mean wind speed in zone 1 is

highest during January (  3 m s$^{-1}$) and in zone 2 during December (> 3 m s$^{-1}$) (Figure 5a). High inter-annual variability is seen in wind speeds along the two zones, with peak wind speed in any one of the months between November and February. Only in certain years, moderate wind (< 3 m s$^{-1}$) coincided with high Chl-a and the correlation coefficient confirms that Chl-a and wind speed are not correlated in zone 1 and 2 (r = 0.09, n=60).

    PAR follows the seasonal cycle of incoming solar radiation (Arnone et al., 1998). PAR is the waveband of light that is used in

photosynthesis, and it is closely correlated with total incoming solar radiation heating the water column. Hence an increase in PAR is accompanied by higher surface temperature and associated with enhanced stratification, which results in reduced mixing and vice versa (Lee et al., 2014). During November to December, low PAR (33-36 E m$^{-2}$ day$^{-1}$) prevailed in the study area, corresponding to low temperature and enhanced mixing, deepening the MLD. Contrarily, when PAR increased after December,

---

[1]Correlation coefficients mentioned in this work are statistically significant at 95% confidence interval



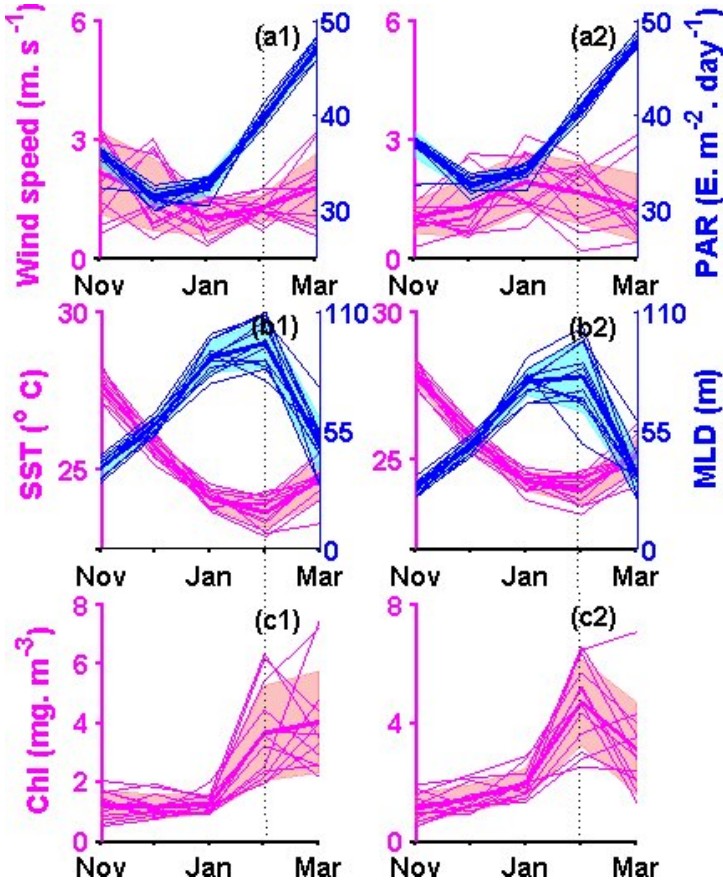

**Figure 5.** Time series of the monthly average concentration of wind speed and PAR (a1 and a2) SST and MLD (b1 and b2) and surface Chl-a (c1 and c2), in zone 1 (left, a1-c1) and zone 2 (right, a2-c2) during the winter period for the years 2002–2013. Thick lines represents mean and the shaded areas the standard deviation. The time series from individual years are shown using thin lines. Vertical dotted lines represent the timing (month) of peak algae blooms.

surface temperature started increasing and mixing was reduced. However, there is a one-month time lag between the onset of increasing PAR and the onset of increasing SST. The increase also coincides with a reduction of the MLD. This is due to the high heat capacity of water and the large amount of energy required to heat the water column when the MLD is deep. Low SST and deep MLD favours increased nutrient supply to the euphotic zone (Madhupratap et al., 1996; Wiggert et al., 2000)

5 and hence production increases in these zones by February. These transitions in terms of a reduction in SST and peak MLD initialise algal blooms. Hence, PAR influences production also indirectly, by affecting the stratification that controls nutrient availability. There is stronger correlation between Chl-a concentration and PAR in zone 1 ($r = 0.69$, $n=60$) and compared to zone 2 ($r = 0.49$, $n=60$).

Peak Chl-a concentrations occurring for the month of February, coincided with lowest SST ($< 25$ °C) and deepest MLD

10 (90-110 m). Nitrate is high at 100 m depth ($> 12$ µmol l$^{-1}$), thus a deepening of the mixed layer beyond 100 m will mix up



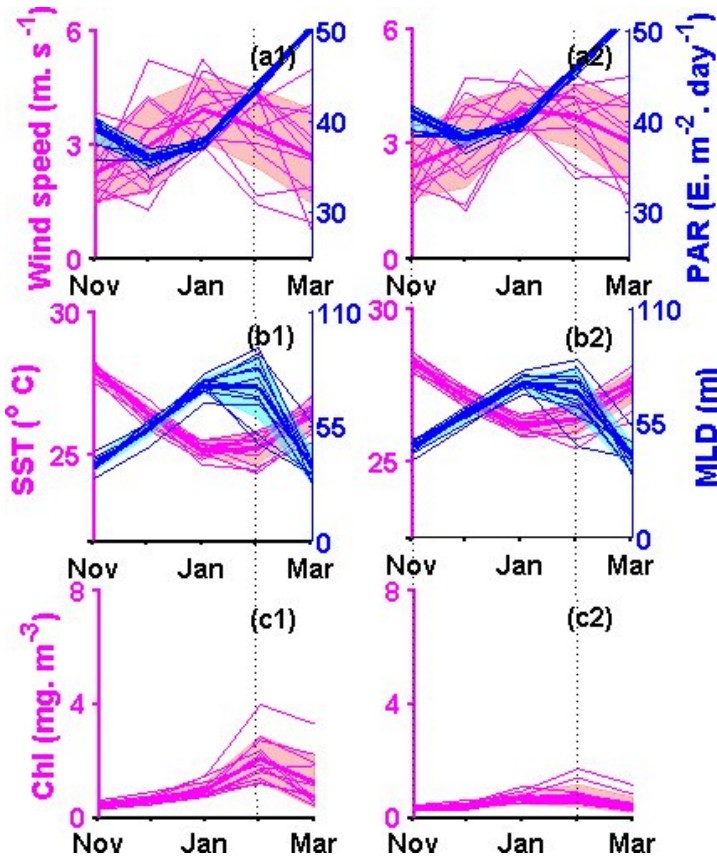

**Figure 6.** Time series of the monthly average concentration of wind speed and PAR (a1 and a2) SST and MLD (b1 and b2) and surface Chl-a (c1 and c2), in zone 3 (left, a1-c1) and zone 4 (right, a2-c2) during the winter period for the years 2002–2013. Thick lines represents mean and the shaded areas the standard deviation. The time series from individual years are shown using thin lines. Vertical dotted lines represent the timing (month) of peak algae blooms.

nutrients towards the surface (Garcia et al., 2013). This suggests that the highest Chl-a concentrations are due to the increase in nutrients by a deepening of the MLD triggered by cooling (Figures 5, 6, 7, 9) in December and January. Wind influence is not strong in these zones. Though the wind speed is relatively low ($<2$ m s$^{-1}$) during most years, certain cases with moderate wind speed ($3$ m s$^{-1}$) are observed in these zones. Moderate wind (5 m s$^{-1}$) occurring during January to February could have enhanced mixing and thus production in zone 2.

## 5.2 The ecological zones in the southern and western Arabian Sea

Chl-a, SST, MLD and PAR in zones 3 and 4 followed similar seasonal patterns of variability as in zones 1 and 2 (Figure 6). However, the range of values are different in these zones, compared with the ecological zones further north. The magnitude of Chl-a concentration in zone 3 is two to three times less than in zones 1 and 2 and Chl-a in zone 4 is about half of the Chl-a



concentration in zone 3. Thus, high differences in Chl-a concentration occur in zones 3 and 4, compared to the first two zones. The inter-annual variability of SST, MLD and PAR are higher in the third and fourth zones compared to the two zones further north. In zone 3, as it is closer to the equator, PAR is 3-4 E m$^{-2}$ day$^{-1}$ higher, SST is 1.5° C warmer and MLD   10 m to 15 m shallower than in zones 1 and 2. Furthermore, PAR in zone 4 is 2-3 E m$^{-2}$ day$^{-1}$ higher, SST is 1.0° C higher and MLD

10 m shallower than in zone 3. The fact that Chl-a concentration in zone 3 is less compared to that of Chl-a in zones 1 and 2 and Chl-a in zone 4 is still lower than that in Zone 3, affirms that variation in Chl-a production in the northern Arabian Sea is strongly related to physical parameters viz. SST, MLD and PAR. SST is an indirect indicator of favourable conditions for algal blooms. Low SSTs can be the result of intensified convection, which will also be manifested by increased mixed layer depth and entrainment of waters rich in nutrients (Morrison et al., 1998). SST in zone 3 is 1.5° C warmer than in zone 1 and

2 and SST in zone 4 is 1° C warmer than in zone 3. Hence, the rate of convection is most likely weak in zone 3 and even weaker in zone 4, compared to zones 1 and 2. This indicates that phytoplankton production in these zones could be linked to the convectional strength and increase in nutrients and that the production is limited by nutrient in the period before and after the bloom. In zone 3 and zone 4 the inverse correlation between SST and Chl-a is stronger, compared to zone 1 and 2, with r = 0.62 and -0.70, respectively.

The availability of nutrients is the prime factor influencing production. A deepening of the mixed layer will increase nutrient availability, but the magnitude depends both on the depth of the MLD and on concentration of nutrients below the mixed layer. The relatively shallow MLD and high SST in zones 3 and 4, compared with zones 1 and 2, suggests low transport of nutrients into mixed layer in zones 3 and 4, compared to the other two zones. MLD and Chl-a productivity in zones 3 and 4 are correlated (r = 0.50, n=60 and r = 0.56, n=60, respectively). The indirect influence of solar radiation in maintaining SST and MLD and

thus nutrient availability is evident from higher PAR, higher SST and more shallow MLD values in zones 3 and 4, compared with zones 1 and 2. Hence, a direct dependence of SST cooling and deepening of the MLD and indirect dependence of PAR with the primary production is evident in the first four zones. In zone 3, a weak correlation exists between PAR and Chl-a (r = 0.41, n=60) and in zone 4, these two parameters are not correlated at all. The increasing wind speed pattern prevalent during winter indicates that wind mixing could be the prime factor governing the ecological dynamics in this zone during winter.

Chl-a has a weak positive correlation with the wind speed in zone 3 (r = 0.30, n=60) and moderately correlated in zone 4 (r = 0.47, n=60). Relatively warm surface and shallow mixed layer in zone 4 indicate weak convective overturning (Naqvi et al., 2006). Hence, wind induced mixing in this zone can influence production.

## 5.3   The ecological zones in the coastal and continental shelf waters

Elevated values of Chl-a (> 2.5 mg m$^{-3}$) persist in zones 5 and 6 throughout the winter season, with high levels of variability.

This suggests that the dynamics in the coastal and continental shelf zones 5 and 6 are more complex than open ocean waters (zones 1, 2, 3 and 4). Chl-a in zone 5 shows significant inter-annual variability for the winter period with its peak value during February. In zone 6 there is low variability of Chl-a for the winter period in January, while the range of variability is high for both December and February months. In zones 5 and 6, MLD maxima and SST minima occurred either during January or February in each of the studied years (Figure 7b). During January and February, in zone 5 MLD varied between 70 to 80 m,





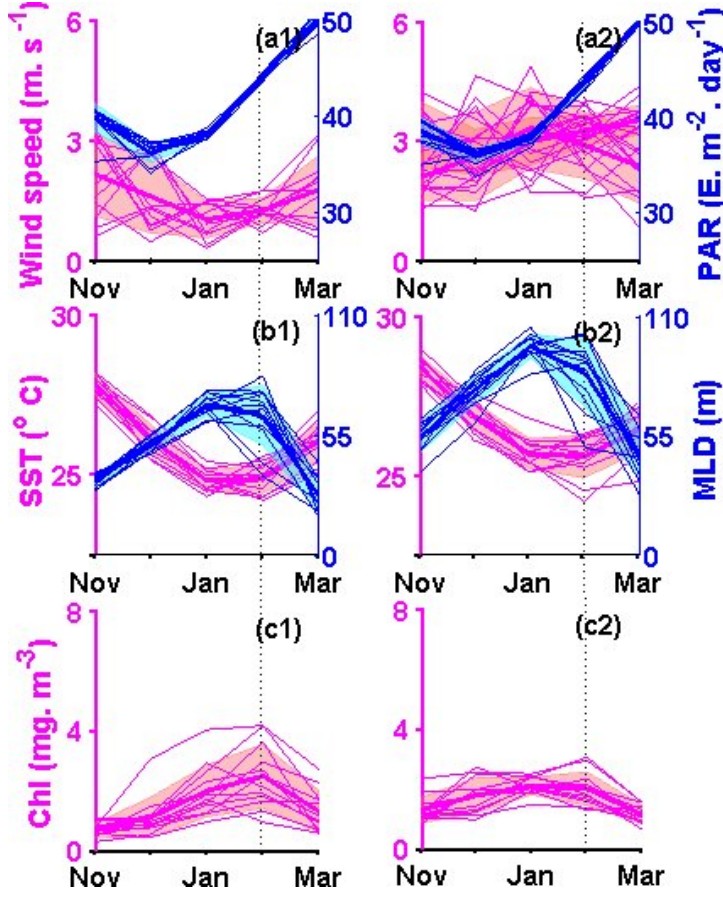

**Figure 7.** Time series of the monthly average concentration of alongshore wind speed and PAR (a1 and a2) SST and MLD (b1 and b2) and surface Chl-a (c1 and c2), in zone 5 (left, a1-c1) and zone 6 (right, a2-c2) during the winter period for the years 2002–2013. Thick lines represents mean and the shaded areas the standard deviation. The time series from individual years are shown using thin lines. Vertical dotted lines represent the timing (month) of peak algae blooms.

and in zone 6, MLD varied between 80 to 100 m. A comparison between MLD values in zones 5 and 6 with those in zones 1 and 2 shows that MLD is shallower in zone 5, whereas the variability of MLD in zone 6 is comparable to that in the first two zones. MLD variability for the winter is consistent with Longhurst's observations, however, the range of MLD in Longhurst is smaller (range from 40 to 70 m) compared to present study (40-110 m) for the winter period. Additionally, SST is higher

5    during January – February in zones 5 and 6 (>24.5 °C), compared to SST values in zones 1 and 2 (23.5-24.5°C). The correlation coefficient between MLD and Chl-a is lower in zone 5 (r = 0.53, n=60) than in zone 6 (r = 0.69, n=60). On the other hand, the inverse relation between SST and Chl-a concentration is higher in zone 5 (r = -0.64, n=60) compared to zone 6 (r = -0.54, n=60). PAR ranges for December and January in zones 5 and 6 are almost equal to PAR ranges in zones 3 and 4 (36-38 E m$^{-2}$ day$^{-1}$) and are higher compared to zones 1 and 2 (30-34 E m$^{-2}$ day$^{-1}$) for the same period. A weak inverse correlation (r =





**Table 1.** Multiple-linear regression analysis

| Zone | Multiple-linear regression equation | r |
|------|-------------------------------------|-----|
| Zone 1 | Chl-$a$ = 0.18 - 0.23 SST+ 0.47 PAR- 0.04 MLD- 0.01 WND | 0.73 |
| Zone 2 | Chl-$a$ = 0.18 - 0.24 SST+ 0.41 PAR+ 0.27 MLD- 0.13 WND | 0.71 |
| Zone 3 | Chl-$a$ = 0.10 - 0.09 SST+ 0.39 PAR+ 0.40 MLD- 0.03 WND | 0.75 |
| Zone 4 | Chl-$a$ = 0.26 - 0.37 SST+ 0.09 PAR+ 0.10 MLD+ 0.09 WND | 0.71 |
| Zone 5 | Chl-$a$ = 0.10 - 0.21 SST+ 0.28 PAR+ 0.44 MLD- 0.06 WND | 0.69 |
| Zone 6 | Chl-$a$ = 0.58 - 0.36 SST- 0.19 PAR+ 0.28 MLD- 0.11 WND | 0.73 |

-0.27, n=60) exists between PAR and Chl-a in zone 6, while in zone 5 these parameters are not correlated (r = 0.12, n=60). For zone 5, wind and Chl-a production are weakly correlated (r =0.30, n=60), while in zone 6, these parameters are not correlated (r = -0.09, n=60).

In zone 5, low SST prevails which is an indicator of strong convective activity. Again, in zone 6, high SST coincides with deep MLD, and strong wind. Wind is reported has the one of the main forcing factor in the INWM by Longhurst (2006), which is consistent with our present study. Comparative warm surface water indicates convective overturning is weak, in addition the presence of strong wind suggests production in this zone could be controlled by upwelling induced by wind. Discharge from the rivers Narmada and Tapi are an additional source of nutrient supply to this region. Hence, the highly varying production in this area can be attributed to this enhanced nutrient supply (Singh and Ramesh, 2011). In addition, in zone 6 it could also be attributed to higher rates of nitrogen fixation (Gandhi et al., 2011; Singh et al., 2012).

## 5.4 Multiple Linear Regression Analysis

Multiple linear regression analysis is carried out to understand the combined effect of all chosen environmental parameters on Chl-a production. Multiple linear regression (MLR) is performed here on normalised values of selected parameters, such that individual values are subtracted from minimum value of observation and then divided by the range of observation. MLR equations with r value are tabulated in table 1. MLR equations for the six zones done in this work are found to be statistically significant and are carried out using 60 data points. In zone 5 and 6, WND represents alongshore wind component.

In general, MLR analysis confirms that production is controlled by surface cooling, enhancement of PAR, deepening of MLD. However, the dependence of each of these variables varies differently within each zone. A negative impact of wind is observed in first three zones and a positive influence of wind in last three zones. As for zone 5 and 6, alongshore wind component are considered, southward component enhances production in these case. However, the negative wind speed coefficient observed for first three cases may be due to the fact that the time span considered in this work is monthly and the effect of wind occurs much less than this time scale. In the first zone, for each 1 % increase in light availability as well as each 1 % cooling enhances Chl-a production by 0.47 mg m$^{-3}$ and 0.23 mg m$^{-3}$ respectively. In second zone PAR (for each 1 % increase production enhances by 0.41 mg m$^{-3}$) has major influence on production, followed by MLD and SST. For the third zone,




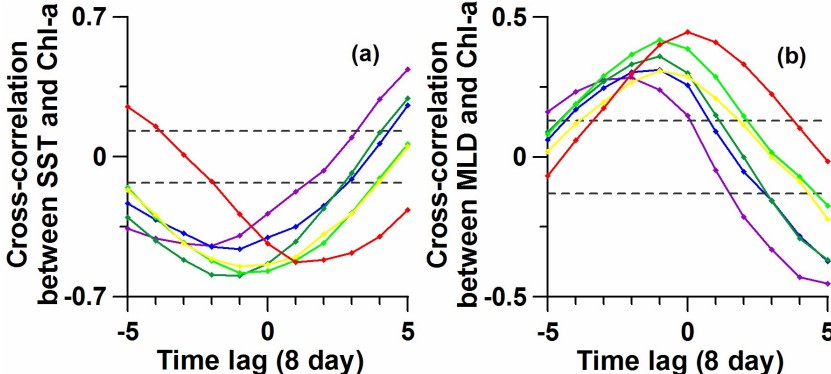

**Figure 8.** Cross-correlation of (a) SST and (b) MLD with Chl-a in six zones. Grey dashed horizontal line represents the 99 % confidence interval. Each tic on X-axis represents an 8-day period.

MLD and PAR has the highest influence on Chl-a. Cooling has major influence in fourth zone. While in fifth zone, MLD, PAR and SST has major control on production, similar to zone 2, however, the rate of enhancement differs between these zones. In zone 6, too cooling has major influence, which is followed by PAR decrease and MLD deepening. An inverse relation of PAR to Chl-a is observed in this zone. It is to be mentioned here that apart from other five zones, two peak are observed on Chl-a

5 maxima, SST minima and deepening of MLD with initial peak during December. PAR has its minima during this initial bloom month and increasing trend corresponding to the second peak of Chl-a i.e., during its minima and maxima value of PAR winter cycles bloom occurs. This implies PAR is not a limiting factor for production in this zone.

## 6 Time lag between SST and MLD variability to peak algae bloom

It is evident from the above analysis that Chl-a production depends strongly on cooling intensity (variability of SST) and MLD

development. To quantify the eventual lag between SST minimum and MLD maximum to Chl-a maximum, the time-lagged correlations of each of these parameters with Chl-a are calculated (Figure 8). As, these parameters induce algal blooms at much shorter time scale than a month, these analyses are carried out using 8-day composite data. Cross-correlation analysis shows (Figure 8 and Table 2) that zones 1 to 5 reveal a strong and significant ($p<0.01$) correlation between Chl-a and SST occurs with lag of -3 to -1 time interval (a scale = 8 days), i.e., dip in SST occurs before the peak in Chl-a. A similar situation is observed

for MLD but the lag is shortened by one time step i.e., 8 days. In zone 6, SST maximum is observed 1 time step later than Chl-a maximum (lag = 8 day) and MLD peaks simultaneously with Chl-a (lag = 0). These observations enable us to put forth a hypothesis that the prevailing cool conditions must have enhanced mixing in the study area, which led to increased algae production.





**Table 2.** Lag between peak Chl-*a*, SST and MLD in 8-day intervals.

| Zone number | Lag between Chl-*a* and SST | Lag between Chl-*a* and MLD |
|:---:|:---:|:---:|
| 1 | -3 | -2 |
| 2 | -1 | -1 |
| 3 | -2 | -1 |
| 4 | -1 | -1 |
| 5 | -1 | -1 |
| 6 | 1 | 0 |

## 7 Impact of nutrients and iron on Chl-a production based on the analysis of climatological nutrient and dust optical thickness

Time lag between Chl-a maxima and MLD maxima suggests that enhanced nutrient availability in the water column due to a deepening of the mixed layer could lead to increased primary productivity. However, the time lags between MLD and Chl-a

varied for the six zones, implying productivity is not only dependent on nutrient availability, but also on other environmental variables (Table 2). Naqvi et al. (2010) reported that iron limits the marine productivity along the Oman coast, which corresponds to the northwestern part of the study area. Similarly, Banerjee and Kumar (2014) have also reported marine production limited by availability of iron in central Arabian Sea. Hence iron supply to the ocean surface have been analysed using the Dust Optical Thickness (DOT), where high DOT indicates more iron deposition from the atmosphere. The temporal variability

of nitrate in the mixed layer and DOT from the atmosphere is compared with the Chl-a variability in each ecological zone (Figure 9). Nitrate and DOT show significantly different patterns of seasonal variability in each zone. Wiggert et al. (2006) parameterized nitrogen half saturation constants in the northern Indian Ocean to 0.4 µmol $l^{-1}$ for small phytoplankton and 0.8 µmol $l^{-1}$ for large phytoplankton. The climatological data show that nitrate < 0.8 µmol $l^{-1}$ was observed only in March (Zone 1) and November (Zone 6), which shows that usually nitrate is not a limiting factor.

High amounts of nitrate can contribute to the production of large algal blooms (Pondaven et al., 2000; Wiggert et al., 2006). These can also be harmful, which are abundant in the eastern Arabian Sea (Singh et al., 2014). However, our observations indicate that higher nitrate does not always correspond to elevated Chl-a concentrations. For example, during the months of December and January high nitrate availability (> 2 µmol $l^{-1}$) prevails for zones 1-5, while biological activity is moderate (Chl-a < 3.0 mg $m^{-3}$), suggesting that additional variables play a role in determining primary production. The fact that, during

each of the algal blooms (Chl-a > 1.5 mg $m^{-3}$) both nitrate (> 3 µmol $l^{-1}$) and DOT (> 0.11) had high values, confirms that the co-occurrence of high concentrations of these two nutrients is necessary to enhance primary production. Interestingly, Chl-a and DOT followed similar pattern of variability from January to March for zones 1-3 and 5. The fact that Chl-a follows a similar temporal pattern as DOT in zones 1, 2, 3 and 5, strongly indicates that iron is a limiting factor for productivity in these zones. This result is in agreement with Wiggert et al. (2006) and Naqvi et al. (2010), who show that iron limits production in



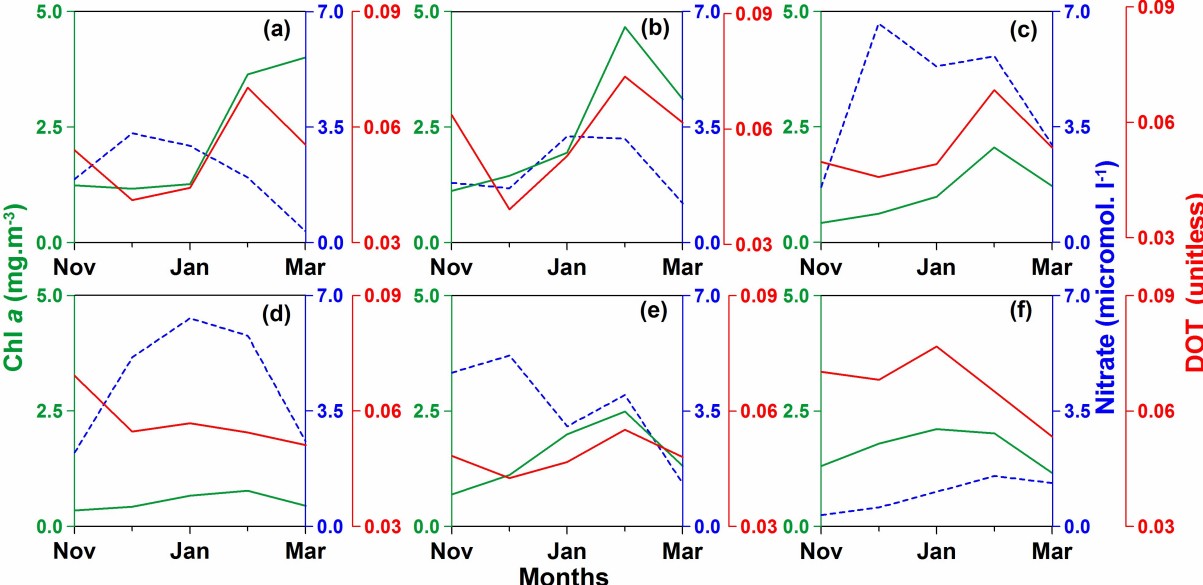

**Figure 9.** Averaged variability of surface Chl-a, nitrate and DOT in six ecological zones. Viewports (a), (b), (c), (d), (e) and (f) represents variability along first, second, third, fourth, fifth and sixth zones, respectively.

the northern and northwestern parts of the Arabian Sea. In zones 4 and 6, which lie to the south and southeast, the relationship between iron and Chl-a is not evident. Aerosol in the south and east has much lower iron content, compared to the western part (Kaufman et al., 2005; Wiggert et al., 2006). To quantitatively analyse the impact of atmospherically deposited iron in the study area, comprehensive in-situ measurements of the iron content at the sea surface are required, and these are presently not
available.

## 8    Summary and Conclusions

In this study a statistical objective zoning methodology was applied to remotely sensed Chl-a data for the northern Arabian Sea and eight homogeneous ecological zones were delineated. In six of these zones Chl-a variability is studied in relation to physical-chemical parameters. The identified six ecological zones give a more comprehensive picture of the variability of
marine ecosystems during winter in the Arabian Sea than the Longhurst classification in two provinces for the entire northern Arabian Sea (Longhurst, 2006). The Chl-a variability followed a semi-cyclic pattern during the winter period, with the mean of peak observations for the study period observed during February in zones 1 to 5 (Figure 4). For zone 6, there is no distinct peak value between December to February. Zones 1 and 2 in the northern part of the Arabian Sea were highly productive (Chl-a values ranging from 1 to 7 mg m$^{-3}$), while zone 3 (1 - 4 mg m$^{-3}$) and zone 4 (1 - 2 mg m$^{-3}$) were found to be less productive,
i.e. a north-south gradient in the phytoplankton productivity is observed. Contrary to the open ocean zones, the coastal and continental shelf water zones, zone 5 and zone 6, have high levels of variability with elevated Chl-a values throughout winter





(> 2.5 mg m$^{-3}$). In addition, the inter-annual variability for the winter season is well captured in the present study and is not seen in Longhurst's case. This is because in the present analysis delineation is done considering winter period alone, while in case of Longhurst the annual variation is considered for delineation. Moreover, this study is assessed for eleven years, while Longhurst's is for about four and a half years (Longhurst, 2006).

The increased amount of Chl-a production in the open ocean zones are found to be directly related to sea surface temperature variability (ie. cooling) and the deepening of the mixed layer. Similar inverse relation between productivity and SST is observed in the Indian Ocean by Singh and Ramesh (2015). PAR was found to have an indirect influence on primary production in the study area. PAR increases surface temperature as a result mixing gets reduced and vice versa. During the month of January, PAR enhancement coincides with SST reduction / MLD deepening. This coincidence occurs with a one month lag between

PAR increase and SST reduction / MLD deepening. SST reduction and MLD deepening increases nutrient supply to the mixed layer thus enhancing production. Production in zone 6 was found to be more complex with influence of wind, river discharges from Narmada and Tapi river (Figure 4) and also attributed to higher rates of nitrogen fixation (Singh and Ramesh, 2011; Singh et al., 2012; Gandhi et al., 2011). MLR analysis confirms production is controlled by surface cooling, increase in PAR and deepening of MLD, with the varying dependence for each of these variables within each ecological zone. To understand wind

dependence on Chl-a, much shorter time scale is required, with variability on scales less than a month.

    Low SST and high MLD prior to onset of the winter algal blooms suggest that nutrient supply from below the thermocline could trigger their onset. However, the temporal variability of the algal blooms in the identified ecological zones could not be explained exclusively in terms of nitrate supply or iron availability, as resolved in the variations in DOT. The combined analysis of DOT and nitrate suggests that the variability of the algae blooms depend on both sources in these zones. The variability of

Chl-a in the northern and northwestern parts of the Arabian Sea is correlated strongly with the atmospheric deposition of iron from January to March.

    The satellite based Chl-a concentration utilized in this work is a proxy of marine primary production, and the results obtained in this work are consistent with those of Singh and Ramesh (2015). Their paper states nutrients and solar radiation are predictors that can explain most of the variability in the marine productivity, and observed a strong inverse relation of primary production

with SST.

    This study provides a more comprehensive understanding of the environmental factors controlling the spatio-temporal variability of the marine chlorophyll a concentration in the northern Arabian Sea during winter conditions. Considering the availability of long time series of high-resolution satellite ocean colour data and biogeochemical numerical ocean models today, this study is timely. Additionally, this study reveals the need for better understanding of factors controlling the marine primary

productivity in other coastal upwelling zones. The north Arabian Sea is not well sampled and more in-situ observations are needed in order to validate remote sensing products and initialise numerical models and establish more reliable databases. Biogeographical studies of the lower trophic level of the marine ecosystem, such as this one, could be used to design new sampling programs and strategies.



*Data availability.* Chl-a, SST, PAR, wind and nitrate climatology used in this work are publicly availbale. However, MLD data used is not publicly available

## Appendix A

### A1 Various combinations of PC and cluster number for performing zoning

As the first three PC's account for 97% of the total Chl-a variability in the study area, it is compulsory to consider at least the first three PC's for zoning. Hence, in the present study various possible combinations of PC's viz., first three PC's, first four PC's and first five PC's are selected to map Chl-a zones. Varying complex coastal dynamics including high Chl-a along the Arabian Peninsula coast; river discharge from Indus river along Pakistan and western Indian coast and high sediment distribution and river discharge from Narmada and Tapi rivers along Gujarat coast suggests at least three ecosystem zones

along the coastal regions (Chandramohan and Nayak, 1991; Singh and Ramesh, 2011). High-saline waters in the Persian Gulf have different dynamics compared to the rest of the study area, suggesting at least one zone in the Persian Gulf. Furthermore, in the open ocean at least two zones are proposed: one in the north and another in the southern sectors (Gomes et al., 2008). Thus, based on the dynamics in the area, at least six distinct zones are identified in the study area. Initial preliminary images showed cluster number nine and above, has insufficient clustering and therefore, cluster number (c) chosen here, is six to eight

(Figure A1).

Various combinations of PC's and clusters are carried out using 3 to 5 PC's and 6 to 8 clusters. The number of PC's selected is hereafter suffixed using letter 'pc' and number of cluster by 'c'. The selected nine combinations of PC's and cluster numbers include (1) 3pc 6c, (2) 4pc 6c, (3) 5 pc 6c, (4) 3pc 7c, (5) 4pc 7c, (6) 5pc 7c, (7) 3pc 8c, (8) 4pc 8c and (9) 5pc 8c (Figure A2). In general, zone maps obtained from the nine selected combinations classified Persian Gulf into two zones, offshore area

into four zones and the areas within bathymetry depth 150 m as coastal zones (Figure A3). Open ocean areas are demarcated using blue and green part of the spectrum, while coastal areas by yellow, orange and red colours. In seven out of the nine zone maps ((1) 3pc 6c, (2) 3pc 7c, (3) 4pc 6c, (4) 4pc 7c, (5) 5pc 6c, (6) 5pc 7c and (7) 5pc 8c) the lower portion of Persian waters and southern part of the area as a single zone. However, the dynamics of these two regions are entirely different (Bower et al. 2000; Shankar et al. 2002). This fact suggests, these zone maps does not differentiate surface ocean Chl-a and hence these

zone maps are not selected for the present study. In case of zone map with 3pc 8c combination, red patches, which in general represent coastal region, is not restricted within 150 m depth. Hence, this zone map is also discarded, leaving the zone map with combination 4pc 8c to be selected for the present study.

### A2 Spatial smoothing on the selected zone map

In the selected zone map, overlapping zones are observed especially in the central study area. Zone 1, zone 2, zone 3 and zone

4 and the orange patch along the Oman coast are highly scattered and hence each of these are overlapped one over the other. Simple averaging will remove the characteristics feature along highly overlapping regions and hence smoothing i.e., border



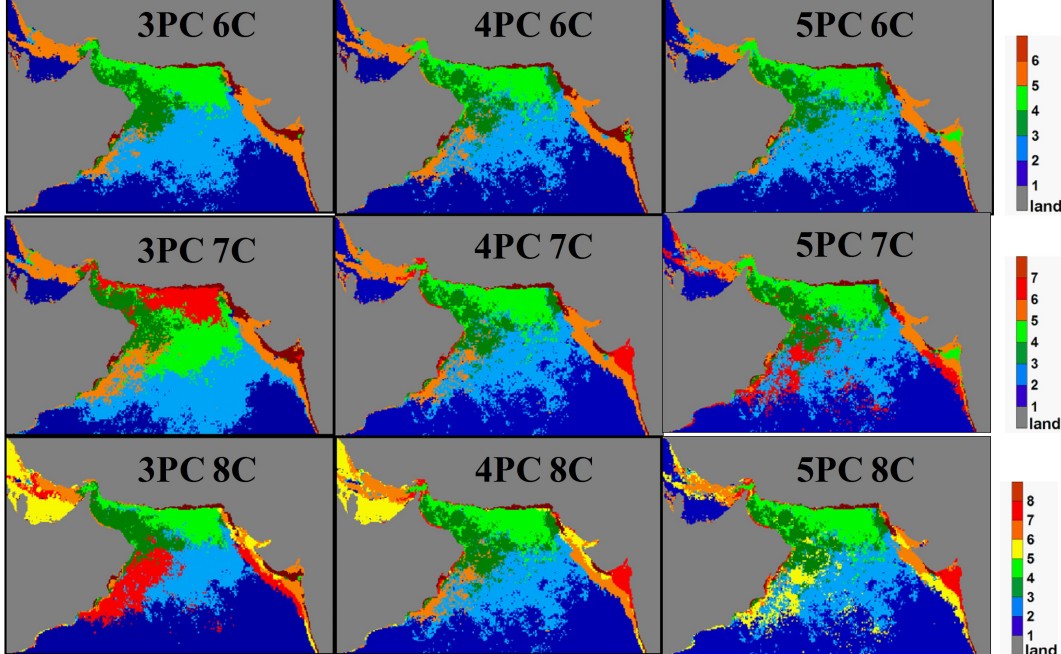

**Figure A1.** Various combinations of PC and CA tried out for achieving better zonation.

of each of these highly scattered zones are identified, before averaging. Smoothing considers an area with 5 x 5 pixel (Figure 4). Each middle pixel is replaced by the zones with major pixel characteristic. Along the coastal area, pixels with more than five consecutively similar values are considered, others are replaced with the main zonation along the area. After smoothing, averaging is applied, around a 3 x 3 pixel area; such that characteristics of pixels with half or more strength are considered,

5    otherwise they are replaced by the main zone.

*Author contributions.*  SS and AK conceived the idea and developed the methodology. SS collected, analyzed and interpreted the data. SS, AS, BB, NM, LP and AK contributed to discussions of the findings. SS wrote the manuscript with contributions from AS, LP and BB.

*Acknowledgements.*  This work was carried out under the project INDO-European Research Facilities for Studies on MARine Ecosystem and
10   CLIMate in India (INDO-MARECLIM) supported by the European Commission under the Seventh Framework programme (INCO-LAB),
     GA# 295092. The study has been conducted in cooperation between scientists at the Nansen Centers in India, Norway and South Africa,



supported by the basic funding at the Nansen Center in Bergen. Authors are grateful to NASA in making Ocean colour available, NODC for nitrate climatology and ECMWF for ERA-Interim data. The development of the regional HYCOM used in this study was jointly supported by the South African National Research Foundation and a grant for computer time from the Norwegian Program for supercomputing (NOTUR project number nn2993k). Prof. (Dr.) Trevor Platt is acknowledged for constructive review of the manuscript for this article.



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
