# Peer review of "Delineation of marine ecosystem zones in the northern Arabian Sea during winter"

_Biogeosciences, 2017_

## Referee Comment (RC1) · Anonymous Referee #1 · 24 Jul 2017

Authors have analysed satellite chlorophyll data to delineate ecological zones in the Arabian Sea as done by Longhurst in the global oceans. With the use several other archival data sets and using principal component analysis, they have categorized the Arabian Sea into six zones. Their analysis is interesting but I have a few comments that can improve the manuscript.

Major comments:

1. On what basis, principal component analysis based six ecological zones were divided into two Longhurst provinces? It should be elaborated in the section 4.1. 2. Because of the lack of satellite data during the monsoon seasons, authors have considered only winter data. We know that the Arabian Sea is most productive during summer. Authors should discuss how ignoring monsoon would impact the delineation

of ecological zones?

3. It appears that only surface chlorophyll values were used. But we know that ocean has a deep chlorophyll maximum (DCM). The justification provided on pages 2-3 (line 34, line 1-2; "the fact that during............content") is not correct. There are numerous studies showing DCM in the Arabian Sea. Authors have covered almost entire Arabian Sea, and it is not possible to have a weak DCM everywhere. Authors should explain how this analysis would be affected by excluding the most productive parts of the ocean?

4. Provide a climatological data based Chlorophyll image as Fig. 4 (c). It would help to see whether chlorophyll content are drastically different in these six zones (particularly for sentence on Page 10, lines 17-18)

Minor comments:

Title should be revised as "Delineation of marine ecosystem zones in the northern Arabian Sea during winter"

Page 3, lines 9-12 can be deleted as they do not provide any info

Page 4, lines 12-13, same font should be used for variables

Page 4, line 19: (Levitus, 1982) has proposed density criteria to estimate MLD which is used widely (Gardner et al., 1995) and a better criterion than temperature.

Page 5, line 18: I should be in italic, in fact all the variables should be made italic throughout

Page 7, line 4: Oman is an upwelling so how could it be oligotrophic (Wyrtki, 1973)

Page 8, line 2: (Naqvi et al., 2010) have not done sampling off Gujarat and Pakistan

Page 8, line 13: How was coastal Chl a found erroneous?

Page 9, line 2: blows should be replaced by blow

Page 9, line 17: "These coastal areas. . .. . ..winter." Reference is needed.

Page 10, line 2: zone 6 is also an upwelling region (Sudheesh et al., 2016)

Page 10, lines 3-4: "Nutrient supply. . .. . ..zones". Provide reference, perhaps (Singh et al., 2012; Singh and Ramesh, 2011)

Page 10, line 9: parts has should be parts have; and this whole sentence should be revised for grammar

Page 20, lines 5-25: These points should (also) be discussed in the main text (preferably in the discussion).

Page 20, line 13: (Kumar et al., 2017) is another new reference for higher N2 fixation.

It is not clear what Fig. 8 conveys

Units have periods (.) at most places (e.g., Fig. 5, mg.m-3, m.s-1). These periods should be removed throughout the manuscript.

References:

Gardner, W.D., Chung, S.P., Richardson, M.J., Walsh, I.D., 1995. The oceanic mixed-layer pump. Deep Sea Res. Part II Top. Stud. Oceanogr. 42, 757–775.

Kumar, P., Singh, A., Ramesh, R., Nallathambi, T., 2017. N2 Fixation in the Eastern Arabian Sea: Probable Role of Heterotrophic Diazotrophs. Front. Mar. Sci. 4, 80.

Levitus, S., 1982. Climatological atlas of the world ocean. NOAA Prof Pap 13, 1–173.

Naqvi, S., Moffett, J.W., Gauns, M., Narvekar, P., Pratihary, A., Naik, H., Shenoy, D., Jayakumar, D., Goepfert, T.J., Patra, P.K., 2010. The Arabian Sea as a high-nutrient, low-chlorophyll region during the late Southwest Monsoon. Biogeosciences.

Singh, A., Gandhi, N., Ramesh, R., 2012. Contribution of atmospheric nitrogen deposition to new production in the nitrogen limited photic zone of the northern Indian Ocean. J. Geophys. Res. Oceans 117.

Singh, A., Ramesh, R., 2011. Contribution of riverine dissolved inorganic nitrogen flux to new production in the coastal northern Indian Ocean: An assessment. Int. J. Oceanogr. 2011.

Sudheesh, V., Gupta, G., Sudharma, K., Naik, H., Shenoy, D., Sudhakar, M., Naqvi, S., 2016. Upwelling intensity modulates N2O concentrations over the western Indian shelf. J. Geophys. Res. Oceans.

Wyrtki, K., 1973. Physical oceanography of the Indian Ocean, in: The Biology of the Indian Ocean. Springer, pp. 18–36.

---

## Referee Comment (RC2) · Anonymous Referee #2 · 21 Aug 2017

General comments

Can't say I like this paper. The innovative information established by the authors is meager: all prime features of the phytoplankton field across the north Arabian Sea and their driving processes are known and the present research has not contributed to this knowledge. The authors regard as a major merit of their work a more fine delineation of marine zones in the north Arabian Sea as compared to the ones determined previously by other workers. First of all, the zones established by the authors are readily discernible in the spatial distributions of Chl, and secondly, the established contours of the zones are not proven. This thesis is underpinned by my comments to the text.

The paper composition is also unsatisfactory: instead of partitioning the respective part of the paper into Results and Discussion sections, the authors mixed up the reporting on the results obtained and underpinning of the results' validity. This caused numerous repetitions and unnecessary lengthening of the text.

The authors' English needs to be brushed up

In light of the above and the comments below, I reckon that the paper should be subsumed under the category "major revision".

Specific comments

| paragraph | comment(s) |
|---|---|
| **1. Introduction** | |
| 5 (page 2) | Specify the desert(s); |
| 15 (page 2) | It is insufficient to anticipate: this needs to be proven. |
| 25(page 2) | Why the Chl concentration at 0.5 mg/m3 is used as a criterion? |
| 30 (page 2) | Firstly, Mignot et al. reported solely on Pacific and Mediterranean oligotrophic waters (typically, Chl is significantly under 1 mg/m$^3$). The actual location and degree of "weakness" of deep Chl maxima (DCM) are site-specific. For the locations within the study waters the assertion that DCM did not affect the satellite-borne Chl concentrations needs independent confirmation. The authors write that DCM in the study area is presumingly shallow because of the strong attenuation by surface Chl. A rather strange argument: if the DCM is shallow then it can be "sensed" by the satellite sensor. Besides, the Chl concentrations reported in your study are not likely to affect the downwelling light to a degree of eliminating the DCM optical influence. At least, a Hydrolight experiment can bring certainty in this issue. |
| **2. Data** | |
| 15 (page 3) | There are no assessments of Chl retrieval errors. This issue is essential, because of the above comment, and also because of the optical heterogeneity within the study waters. It is unnecessary to mention that the NASA algorithm used by the authors is valid (and produces really accurate values of Chl concentrations) only for case I waters (i.e. strictly oligotrophic). However, the authors haven't elucidated this issue with regard to the studied waters in view of the impacts produced by the river discharge, and dust fallouts. The observed variations in Chl could arise, inter alia, from the inability of the NASA algorithm to retrieve Chl correctly in those parts of the study sea where waters are not strictly case I waters. In this case the zoning [in essence, based on Chl variations] might be compromised (at least the declared contours of six zones, which are supposed to be the main advantage of the study). That is why the realistic error bars relevant to the study sea are indispensable for all illustrations of Chl concentrations in the selected zones. The issue of retrieval error arises also with respect to other satellite-borne variables used in the study.. |

| | |
|---|---|
| 5 (page 4) | As a matter of fact: the coefficients taken from the literature are not necessarily relevant to the study area, e.g. $f_{du}$, and $AOT_{m\,a}$ (the later was determined by Smirnov et al., for Midway Island in the Pacific, located in waters located far away from the study area; meanwhile, it is known that $AOT_{m\,a}$ depends not only upon the above water surface wind but also on a number of other parameters, that is why there are many parameterizations suggested for specific marine locations). |

| | |
|---|---|
| 20 (page 4) | Please, give the major assessments of MLD simulation errors (results of validation by George et al., 2010). Error bars are indispensably required for all illustrations of MLD variations in the selected zones. |

**4. Objective delineation of ecosystem zones in the northern Arabian Sea**

| | |
|---|---|
| 5(page 7) | If only PC 1-3 are meaningful, why you provide illustrations for PC 4 and PC5 (fig. 2 ). The authors are reporting on the northwestern and southeastern gradients in spatial distributions of PC1 (that is the component that predominantly, , accounts for 97% ofthe spatio-temporal variance in Chl) as one of the important findings. However, this finding could be attained without the PC analysis just by visual examination of the spatial distribution of Chl or/and SST, which is confirmed by the authors themselves. So there is nothing new in this finding. |
| 5 and 10 (page 8) | First, the authors write that PC4 and PC5 are not informative (mostly noise) and then declare that the final delineation into ecological zones was obtained by combining the first 4 PCs. Please, explain. Also, please, explain what you mean saying "based on general Chl pattern in.." |
| 10 (page 8)  15 (page 8) | Please, explain, on the basis of what it was decided that satellite-derived Chl values along coastal and shallow waters were erroneous. Please, explain in the paper what are the reasons to believe that " the physical forcing affecting chl concentration along the two regions is likely to be different" … |
| 5 and 10 (page 9) | The authors write that 1-3 zones (encompassed by Longhurst's ARAB zone) are strong upwelling regions with high Chl in winter time, and then they refer to Longhust who defines the ARAB province as a zone with strong upwelling during summer and strong convective cooling during winter. Obviously, some phrase is required to follow these statements in order to clarify the actual hydrodynamic situation therein. |
| 5 (page 10) | Please, specify 1. what is known about the atmospheric deposition on nitrogen (there is no respective reference), and 2. why this mechanism of nutrient supply acts only in zone 6 (or, at least, is not mentioned with regard to other zones). Also, specify the annual cycle of stream flow of the Narmada and Tapi rivers to support your thesis that nutrient supply from Narmada and Tapi rivers as well as atmospheric deposition of nitrogen enhances marine production in zone 6. This additional information might clarify the authors' statement that in zone 6 "peak Chl-a is observed during January" as opposed to other zones.. |

**5.1  The ecological zones in the northern and most productive part of the Arabian Sea**

| | |
|---|---|
| 15 (page 11) | First, the authors write that the inverse relationship between SST and Chl-a have weak correlation coefficient 1 in zone 1 (r = 0.39, n=60) and zone 2 (r = 0.55, n=60). . Then a bit further: "However, MLD and Chl-a in zone 1 and 2 are moderately correlated (correlation coefficient, r = 0.28)". What are your criteria in this regard? |
| 25 (page 11) | The authors write "Mean wind speed in zone 1 is  highest during January (3 m s–1) and in zone 2 during December (> 3 m s–1) (Figure 5a"). Does fig. 5a collaborates this statement? Further on: "During November to December, low PAR (33-36 E m–2 day–1) prevailed in the study area, corresponding to low temperature and enhanced mixing, deepening the MLD.  But according to fig. 5 in November –December MLD is still rather shallow, especially in November. |

| 5 (page 12) and 5 (page 13) | The fig. captions are poorly written: "Time series of the monthly average concentration of wind speed and PAR (a1 and a2) SST and MLD" |
|---|---|
| 5 (page 16) | Please, comment on your finding that PAR and Chl for zone 5 are not correlated at all, and for zone 6 they are inversely correlated. Also, some interpretational comments are required for the phrase "For zone 5, wind and Chl-a production are weakly correlated (r =0.30, n=60), while in zone 6, these parameters are not correlated (r = -0.09, n=60)" Table1: why the regression equations do not include such variables as MLD, concentration of nitrates nitrates and iron. It would be much better to do so instead of discussing the relations between Chl and the above variables apart from the variables reflected in Table 1. |
| Page 17 | Caption for Fig. 8 lacks the designations of colours |

**7. Impact of nutrients and iron on Chl-a production based on the analysis of climatological nutrient and dust optical thickness**

| 15 (page 13) | Please, give (at least in the Appendix section) the rose of winds in winter in order to let the reader better understand why in some parts of the sea DOT is higher than in the others. It would be good to give alongside it the field of DOT over the study area. |
|---|---|

**8. Summary**

| 10 (page 19) | As was commented above, the reported finding on the north-south gradient in Chl is stale and had been established without any complicated processing procedures. The same comment can be made with regard to the identified number of |
|---|---|
| 5 (page 20) | The reported finding that "The increased amount of Chl-a production in the open ocean zones are found to be directly related to sea surface temperature variability (ie. cooling) and the deepening of the mixed layer " is neither an unknown phenomenon for the study area. |
| 15(page 20) | "The combined analysis of DOT and nitrate suggests that the variability of the algae blooms depend on both sources in these zones. The variability of Chl-a in the northern and northwestern parts of the Arabian Sea is correlated strongly with the atmospheric deposition of iron from January to March"  The two statements appear kind contradictory. |
| 30 (page 20) | It is difficult to agree with the authors' statements that "This study provides a more comprehensive understanding of the environmental factors controlling the spatio-temporal variability of the marine chlorophyll a concentration in the northern Arabian Sea during winter conditions", and further on "Additionally, this study reveals the need for better understanding of factors controlling the marine primary productivity in other coastal upwelling zones". Indeed, to justify/prove the validity of each zone the authors refer to the relevant publications of other workers who investigated in depth the factors and mechanisms controlling the spatio-temporal variability of the marine chlorophyll a concentration. Also, in many studies of the north Arabian Sea the need of further investigations, and more thorough sampling/in situ determinations of physico- and biogeochemical variables. |

---

## Author Comment (AC1) · 7 Oct 2017

Dear Editor,

The authors appreciate the opportunity to improve the manuscript ID bg-2017-285 entitled "Delineation of marine ecosystem zones in the northern Arabian Sea using an objective method" by Saleem Shalin, Annette Samuelsen, Anton Korosov, Nandini Menon, Björn C. Backeberg and Lasse H. Pettersson. We also thank reviewer for critical reviewing and the comments they provided.

All the concerns of referee are addressed and we hope that the after revisions the manuscript will meet the requirements to be published in the Biogeosciences.

Reply by the authors to the referee's comments is herewith attached.

[Figure]

Looking forward to hear from you, With regards,

Dr. Shalin Saleem Research Associate Central Marine Fisheries Research Institute Post Box No. 1603, Ernakulam North P.O., Kochi-682 018. Kerala, India

Please also note the supplement to this comment: https://www.biogeosciences-discuss.net/bg-2017-285/bg-2017-285-AC1-supplement.pdf

**Supplement:**

We appreciate the reviewer for taking the time to make such detailed comments to our manuscript. Special thanks for the suggestions on  $AOT_{ma}$  calculation and the comment on including a figure with wind rose along with DOT. However, we expects the reviewer to be quantitative in referring to the actual publications making a comparable delineation of marine zones for the Arabian Seas, beyond what we regard as the state of the art and have quoted and based our publication on.

**General comments:** "Can't say I like this paper. The innovative information established by the authors is meager: all prime features of the phytoplankton field across the north Arabian Sea and their driving processes are known and the present research has not contributed to this knowledge. The authors regard as a major merit of their work a more fine delineation of marine zones in the north Arabian Sea as compared to the ones determined previously by other workers. First of all, the zones established by the authors are readily discernible in the spatial distributions of Chl, and secondly, the established contours of the zones are not proven."

Authors' comments: In order to "prove" the robustness of our delineation of the identified zones a new figure representing the seasonal average of Chl-a over the winter period (Nov-March) is included as Figure R1, which reveals distinct Chl-a characteristics for each of the identified ecological zones. Our objective classification based on winter average of Chl-a values from eleven winter seasons takes into account both spatial and temporal information. To say that the same result could be obtained by the authors by looking at the spatial distribution is highly uncertain and the result would probably depend both on the person doing the subjective analysis and how the data was presented in terms of colormap etc.. In the initial manuscript itself, the authors have compared Chl-a variability in six obtained zones with the well-accepted biogeographic classification of Longhurst falling in the selected area. As our study has utilised Chl-a concentration obtained from satellite sensors which has about 100 times finer spatial resolution used by Longhurst for regional mapping for classifying ecological zones in the northern Arabian Sea, this regional classification could delineate the spatial Chl-a variability better with more detailed regional information than obtained from Longhurst's classification. The objectivity of the methods used and the increased amount of information in modern ocean color products are the basis for author's argument about 'finer delineation of marine zones in the north Arabian Sea' is true.

Authors have analysed physical and chemical characteristics within each of the identified marine ecological zones, which relation between cooling deepening and production between six zones. In the analysis section, author has made use of the established knowledge on driving processes of Chl-a processes in the study area based on published information. Our information is based on surface-data and limited number of variables – hence we must utilize previous studies to better understand out results. However, the in-situ observation coverage in the Arabian Sea is lacking both spatially and temporally and the utilized literature base their result on observations from shorter periods compared with the to our study. Such long period of information is very essential for resolving inter-annual variability in the ecosystem characteristics. Our study contributes

understanding of the temporal/spatial variability of phytoplankton and hence, authors disagree with the reviewer's comment on '*The innovative information established by the authors is meager: all prime features of the phytoplankton field across the north Arabian Sea and their driving processes are known and the present research has not contributed to this knowledge*'.

Figure R1: Annual winter climatology (seasonal average Chl-a concentration over the winter period (Nov-March) from 2002 to 2013) of Chl-a revealed from satellite data. The black line indicated the delineated zonal boundaries.

This thesis is underpinned by my comments to the text. The paper composition is also unsatisfactory: instead of partitioning the respective part of the paper into Results and Discussion sections, the authors mixed up the reporting on the results obtained and underpinning of the results' validity. This caused numerous repetitions and unnecessary lengthening of the text. The authors' English needs to be brushed up In light of the above and the comments below, I reckon that the paper should be subsumed under the category "major revision".

Author's agree to restructure the summary and conclusion section. If the editor provide an opportunity this part of write-up will be restructured to fit with the more traditional partitioning of scientific papers and the English will be revised and improved.

Authors do appreciate for the comments on  $AOT_{ma}$  calculation and wind rose. These two comments with several comments on poorly written statement has helped authors to improve the manuscript. However, we disagree on comment about the PCA is very drastic.

**Specific comments**

**1. Specify the desert(s); [5 (page 2)]**

Authors' comment: Arabian desert in the west and Thar desert to the east are the major dust contributing deserts.

**2. It is insufficient to anticipate: this needs to be proven. [15 (page 2)]**

Authors' comment: Agreed. The statement has been rephrased and the following references were included in the text that justifies our argument: Longhurst 1995, Longhurst 1998 and Longhurst 2006; Spalding et al. 2012.

**3. Why the Chl concentration at 0.5 mg/m3 is used as a criterion? [25 (page 2)]**

**Authors' comment:** The concerned statement is a general argument for Chl-a concentration for the study area in an annual cycle (Sarma et al. 2012; Ravichandran et al. 2012). Based on Chl-a monthly climatology for the study area, annual concentration considering all seasons comes around 0.5 mg m-3.

4. Firstly, Mignot et al. reported solely on Pacific and Mediterranean oligotrophic waters (typically, Chl is significantly under 1 mg/m3). The actual location and degree of "weakness" of deep Chl maxima (DCM) are site-specific. For the locations within the study waters the assertion that DCM did not affect the satellite-borne Chl concentrations needs independent confirmation. The authors write that DCM in the study area is presumingly shallow because of the strong attenuation by surface Chl. A rather strange argument: if the DCM is shallow then it can be "sensed" by the satellite sensor. Besides, the Chl concentrations reported in your study are not likely to affect the downwelling light to a degree of eliminating the DCM optical influence. At least, a Hydrolight experiment can bring certainty in this issue. [30 (page 2)]

Authors' comment: We agree that deep Chl maxima are site-specific. However, some regions in the selected area show shallow DCM (24 m) during winter (Al-Niami et al. 2017), and concurrently regions with deeper DCM exist in the study area (Breves et al. 2003; Ravichandran et al. 2012; Kumar 2000). Since, it is clear that DCM is not shallow in the entire study area during winter, the statement 'DCM is shallow during winter' is deleted. However, it is to be mentioned here that in-situ coverage on Arabian Sea is not sufficient to give complete spatial and temporal variability on DCM and hence we have to accept the uncertainty on this issue (Barlow et al. 1996).

5. There are no assessments of Chl retrieval errors. This issue is essential, because of the above comment, and also because of the optical heterogeneity within the study waters. It is unnecessary to mention that the NASA algorithm used by the authors is valid (and produces really accurate values of Chl concentrations) only for case I waters (i.e. strictly oligotrophic). However, the authors haven't elucidated this issue with regard to the studied waters in view of the impacts produced by the river discharge, and dust fallouts. The observed variations in Chl could arise,

inter alia, from the inability of the NASA algorithm to retrieve Chl correctly in those parts of the study sea where waters are not strictly case I waters. In this case the zoning [in essence, based on Chl variations] might be compromised (at least the declared contours of six zones, which are supposed to be the main advantage of the study). That is why the realistic error bars relevant to the study sea are indispensable for all illustrations of Chl concentrations in the selected zones. The issue of retrieval error arises also with respect to other satellite-borne variables used in the study. [15 (page 3)]

**Authors' comment**: The NASA OBPG chlorophyll product that we used does not have values of uncertainties associated with each value of chlorophyll and, therefore, region-wise assessment of errors in the chlorophyll product is not feasible to perform. The validation shows that in oligotrophic waters the algorithm accuracy is quite high:  $r^2 = 0.86$ , RMS = 0.25 mg m-3 (Feldman, 2017; Hu et al., 2012). Large errors are presumably observed in the turbid waters of the Persian Gulf as well as the coastal areas. Our region of interest excluded coastal areas and included only phytoplankton dominated open ocean areas, where the standard algorithm of NASA would work well.

6. As a matter of fact: the coefficients taken from the literature are not necessarily relevant to the study area, e.g. fdu, and AOTm a (the later was determined by Smirnov et al., for Midway Island in the Pacific, located in waters located far away from the study area; meanwhile, it is known that AOTm a depends not only upon the above water surface wind but also on a number of other parameters, that is why there are many parameterizations suggested for specific marine locations). [5 (page 4)]

**Authors' comment:** Thanks for this comment. This question is a valid one, author was not aware of the stated scenario. It is clear now that in the Indian Ocean an exponential relation exist between wind speed and sea salt formation, where as in Pacific this relation is linear. As a result of which now, the author have replaced Smirnov et al. 2003 with Moorthy 1997 to estimate  $AOT_m$ . DOT such obtained is super-imposed in the manuscript figure below (pink line) while red represents DOT as computed with the old formula, as can be seen the values differ, but the temporal evolution is similar in zone 1, 2 and 5 (Figure R2).

---

## Author Comment (AC2) · 7 Oct 2017

Dear Editor,

The authors appreciate the opportunity to improve the manuscript ID bg-2017-285 entitled "Delineation of marine ecosystem zones in the northern Arabian Sea using an objective method" by Saleem Shalin, Annette Samuelsen, Anton Korosov, Nandini Menon, Björn C. Backeberg and Lasse H. Pettersson. We also thank reviewer for critical reviewing and the comments they provided.

All the concerns of referee are addressed and we hope that the after revisions the manuscript will meet the requirements to be published in the Biogeosciences.

Reply by the authors to the referee's comments is herewith attached.

[Figure]

Looking forward to hear from you, With regards,

Dr. Shalin Saleem Research Associate Central Marine Fisheries Research Institute Post Box No. 1603, Ernakulam North P.O., Kochi-682 018. Kerala, India

Please also note the supplement to this comment:
https://www.biogeosciences-discuss.net/bg-2017-285/bg-2017-285-AC2-supplement.pdf

**Supplement:**

Authors thank reviewer for critical reviewing the article. Special thank for the suggestion of including Chlorophyll image based on climatological data. Figure obtained per suggestion verifies the obtained classification is best way of presenting / categorizing Arabian Sea based on Chl-a characteristics for the winter.

**Major comments:**

1. On what basis, principal component analysis based six ecological zones were divided into two Longhurst provinces? It should be elaborated in the section 4.1.

**Authors' comment:** 'Our analysis of 'Chl-a winter variability revealed six distinct ecological zones in the Arabian Sea, which has been compared with the Longhurst biogeographical classification of marine provinces for the study area.' This comparison is done as Longhurst's biogeographic classification is the well accepted one for the world oceans including the Indian Ocean. The statement provide in quotes will be included in the main text.

2. Because of the lack of satellite data during the monsoon seasons, authors have considered only winter data. We know that the Arabian Sea is most productive during summer. Authors should discuss how ignoring monsoon would impact the delineation of ecological zones?

**Authors' comment:** Yes, this is a limitation of our study approach, however the amount of comparable high quality (satellite Earth observation) data coverage in time and space make the statistical analysis and zonal classification robust. Combining satellite Earth observation data with seasonal ship measurements would have been advantageous; however we did not have access to such data to be incorporated in our analysis. The most productive period during summer coincides with persistent cloud cover in the study area (Saha, 1974). Since this work utilises satellite ocean colour data, which are limited by cloud cover, ocean colour data cannot be utilised to study the chosen area during the summer (Martin, 2004). Accordingly our analysis focuses on data from the winter period (Nov-March) in order to examine intra- as well as inter-Chl-a variability in the study area. With this limitation in our study we clearly claim that we have analyzed the intra and inter-winter variability, though the obtained zones would likely be different had it been possible to include the whole year. How it would change is hard to say, but a more prominent signal from the summer northwest Arabian Sea upwelling region is likely.

3. It appears that only surface chlorophyll values were used. But we know that ocean has a deep chlorophyll maximum (DCM). The justification provided on pages 2-3 (line 34, line 1-2; "the fact that during: : :: : :: : :: : :.content") is not correct. There are numerous studies showing DCM in the Arabian Sea. Authors have covered almost entire Arabian Sea, and it is not possible to have a weak DCM everywhere. Authors should explain how this analysis would be affected by excluding the most productive parts of the ocean?

**Authors' comment:** Yes, the present work utilized depth integrated surface chlorophyll values, which is remotely sensed by ocean colour satellite sensors. Reflected radiance is measured by the ocean colour sensor which contains scattered light containing information from ocean recorded

upto the depth where it is no longer reflected back to the surface (i.e., 0.1 photosyntheically available radiation (PAR) depth) (Martin, 2004). Hence remotely sensed Chl-a represents the average of Chl-a concentration from surface upto layer where 0.1 PAR with that of surface is available.

As pointed out, authors do admit the factor that Arabian Sea is not showing weak DCM everywhere during winter (Breves et al. 2003; Revichandran et al. 2012; Prasanakumar 2000). Also, some productivity may be excluded as we are not considering DCM-variability; however with lacking in-situ observations, there is no good way to include the deep layers. Furthermore, the increased chlorophyll at depth is sometimes a result of the phytoplankton having higher Chl/C ratio, to compensate for low light, not necessarily higher biomass. Again we will argue that our homogenous data set with extensive coverage in time and space found a basis for robust statistical analysis as long as the limitations of using satellite EO data are taken into consideration.

4. Provide a climatological data based Chlorophyll image as Fig. 4 (c). It would help to see whether chlorophyll content are drastically different in these six zones (particularly for sentence on Page 10, lines 17-18)

**Authors' comment:** Valid comment – Thanks! A Chl-a image (revealing the seasonal average Chl-a values over the winter period (Nov-March) from 2002 to 2013) is provided as a new figure 1, and it can be placed as 4 (c) as per the suggestion. Also, the following sentence provided in quotes can be included in the manuscript. 'The annual winter climatology (seasonal average Chl-a values over the winter period (Nov-March) from 2002 to 2013) of Chl-a distribution revealed distinct features for each of the identified ecological zones (Figure 4 (c)). Based on the variability of Chl-a concentrations, zone 1 experiences maximum bloom intensity between 1.5 to 9.6 mg m$^{-3}$ with a mean of ~2.6 mg m$^{-3}$ and standard deviation of 0.7 mg.m$^{-3}$. Next to Zone 1, high Chl-a prevails in Zone 2, with a range of 1.4 to 7.0 mg m$^{-3}$ and a mean ~ 2.8 mg m$^{-3}$. Standard deviation observed in Chl-a are same for both zones. Moderate values of Chl-a (1.3 to 1.9 mg m$^{-3}$) are observed in Zone 3, Zone 5 and Zone 6. Though similar range are observed for these three zones, the statistics are different. In zone 3, Chl-a varies between 0.5 to 4.2 mg.m$^{-3}$, with 0.3 mg.m$^{-3}$ deviation. Among coastal zones, zone 6 Chl-a deviation is high (0.8 mg.m$^{-3}$) with a range of 0.9 to 6.8 mg.m$^{-3}$ than for zone 5 (0.5 mg.m$^{-3}$) between 1.0 to 4.3 mg.m$^{-3}$. Minimum value of Chl-a for the winter is observed in zone 4 (0.2 to 1.2 mg.m$^{-3}$), also in this zone least mean (0.6 mg.m$^{-3}$) and standard deviation (0.2 mg.m$^{-3}$) is observed. As Chl-a geo-spatial statistical variation in the study area clearly demarcates different ecological zones, the present classification of ecological zones is best way of presenting / categorizing Arabian Sea based on Chl-a characteristics for the winter.'

[Figure]

Figure 4c: Annual winter climatology (seasonal average Chl-a values over the winter period (Nov-March) from 2002 to 2013) of Chl-a revealed from satellite data. The black line indicated the delineated zonal boundaries.

**Minor comments:**
1. Title should be revised as "Delineation of marine ecosystem zones in the northern Arabian Sea during winter"
**Authors' comment:** The suggested title is appropriate for this work and hence it will be changed according to suggestion.

2. Page 3, lines 9-12 can be deleted as they do not provide any info
**Authors' comment:** The sentence is retained, as this sentence connects various supplementary data used in this work.

3. Page 4, lines 12-13, same font should be used for variables
**Authors' comment:** Suggestion will be incorporated in manuscript.

4. Page 4, line 19: (Levitus, 1982) has proposed density criteria to estimate MLD which is used widely (Gardner et al., 1995) and a better criterion than temperature.
**Authors' comment:** We thank the reviewer for this suggestion. The reason we use a MLD based on temperature criteria is because numerous other studies, including Rao et al. 1989; Rao and Sivakumar, 2000 and Kumar and Narvekar, 2005 used MLD based on temperature criteria in the Indian Ocean basin to study the MLD dynamics in the area. By using the same definition of MLD as these authors allows us to compare our results qualitatively to these previous studies. In addition, Kara et al. 2000 found monthly scale variability of MLD deduced from temperature criteria from Ocean Weather Station data have good agreement with Levitus et al. 1994 and Levitus and Boyer, 1994. Moreover, a comparison of MLD obtained from the HYCOM modeled data using both temperature as well as density criteria's with the Argo datasets available for winter are carried out. A total of 6256 points are collocated for winter for the entire study area for the comparison. MLD calculated from density criteria have more RMSD value and error percentage (RMSD: 36 m and an error of 68 %) compared with that derived from temperature criteria using $1^{\circ}$ C, $0.5^{\circ}$ C and $0.2^{\circ}$ C (RMSD: 20 m and an error of 28 %). This analysis showed better MLD derivation is with temperature criteria. Hence, a second analysis based on different

temperature based MLD criteria ($1^o$, $0.5^o$ and $0.2^o$ drop from that at surface) with the Chl-a in the six zones were carried out. From this analysis, it was found that MLD calculated using temperature criteria ($1^o$C degree) could explain the Chl-a pattern in each of the six selected zones more accurately than those computed using other temperature values. This is the reason for include temperature based MLD in the present work.

5. Page 5, line 18: I should be in italic, in fact all the variables should be made italic throughout
**Authors' comment:** Suggestion will be incorporated in manuscript

6. Page 7, line 4: Oman is an upwelling so how could it be oligotrophic (Wyrtki, 1973)
**Authors' comment:** Author accepted the mistake, it is to be replaced as mesotropic. However, explanation of Principal Component Analysis (the corresponding paragraph) is rewritten based on periodicity of Principal Components and hence this sentence will be removed from manuscript.

7. Page 8, line 2: (Naqvi et al., 2010) have not done sampling off Gujarat and Pakistan
**Authors' comment:** Author apologies for the mistake. The reference Naqvi et al. 2010 should be replaced by Sarma et al. 2012. However, this sentence too coincides with Principal Component Analysis which is rewritten and this sentence will be removed.

8. Page 8, line 13: How was coastal Chl a found erroneous?
**Authors' comment:** We used the NASA OBPG chlorophyll-a product derived with the OC4 band ration algorithms performing well only in Case-I waters (see also reply to question 5) [O'Reily et al., 1998]. Since the coastal zone is loaded with turbid waters (due to river inflow or resuspension) and may be optically shallow, the OC4 algorithm is not applicable and these zones were excluded from the analysis.

9. Page 9, line 2: blows should be replaced by blow
**Authors' comment:** Suggestion will be incorporated in manuscript

10. Page 9, line 17: "These coastal areas: : :: : :.winter." Reference is needed.
**Authors' comment:** Reference viz. Kumar and Prasad, 1994; Kumar et al. 2001 will be included in the manuscript.

11. Page 10, line 2: zone 6 is also an upwelling region (Sudheesh et al., 2016)
**Authors' comment:** Two reference will be added including Sudheesh et al. 2016 and Shalin and Sanilkumar 2014 in the manuscript

12. Page 10, lines 3-4: "Nutrient supply: : :: : :.zones". Provide reference, perhaps (Singh et al., 2012; Singh and Ramesh, 2011)
**Authors' comment:** Suggestion will be incorporated in manuscript

13. Page 10, line 9: parts has should be parts have; and this whole sentence should be revised for grammar

**Authors' comment:** Suggestion is incorporated will be included by incorporating following sentence given in quotes in the manuscript. 'However, we have identified high Chl-a concentration ($>0.5$ mg m$^{-3}$) in the entire study area, with significant differences between various parts, particularly higher values to the waters closer to the coast.'

14. Page 20, lines 5-25: These points should (also) be discussed in the main text (preferably in the discussion).

**Authors' comment:** Suggestion of included lines 5 to 25 from summary to discussion section will be incorporated in the manuscript.

15. Page 20, line 13: (Kumar et al., 2017) is another new reference for higher N2 fixation.

**Authors' comment:** Suggestion will be incorporated in the manuscript.

16. It is not clear what Fig. 8 conveys

**Authors' comment:** Sorry for the oversight mistake for not including colour description. Sentence provided in quotes will be included in the Figure caption. "Zones 1 to 6 are represented by violet, blue, green, light green, yellow and red lines respectively."

17. Units have periods (.) at most places (e.g., Fig. 5, mg.m-3, m.s-1). These periods should be removed throughout the manuscript.

**Authors' comment:** Suggestion will be incorporated in the manuscript.

**References:**

1. Breves W., Reuter R, N. Delling and W Michaelis 2003 Fluorophores in the Arabian Sea and their relationship to hydrographic conditions Ocean Dynamics(2003) 53: 73–85, DOI 10.1007/s10236-003-0025-z
2. Gardner, W.D., Chung, S.P., Richardson, M.J., Walsh, I.D., 1995. The oceanic mixedlayer pump. Deep Sea Res. Part II Top. Stud. Oceanogr. 42, 757–775.
3. Kara, A. B., P. A. Rochford, and H. E. Hurlburt (2000), An optimal definition for ocean mixed layer depth, J. Geophys. Res., 105(C7), 16803–16821, doi:10.1029/2000JC900072.
4. Kara, A.B., Rochford, P.A., Hurlbutt, H.E., 2000. Mixed layer depth variability and barrier layer formation over the North Pacific Ocean. J. Geophys. Res. 105 (C7),
5. Kumar S P, M Madhupratap, M D Kumar, M Gauns, P M Muraleedharan, V V S S Sarma and S N De Souza, 2000. Physical control of primary productivity on a seasonal scale in central and eastern Arabian Sea Proc. Indian Acad. Sci. (Earth Planet. Sci.), 109, No. 4, pp. 433-441
6. Kumar S. P. and Prasad T. G. Winter cooling in the northern Arabian Sea, Current Science, 71(11), 834-841.
7. Kumar, P., Singh, A., Ramesh, R., Nallathambi, T., 2017. N2 Fixation in the Eastern Arabian Sea: Probable Role of Heterotrophic Diazotrophs. Front. Mar. Sci. 4, 80.
8. Kumar, S. P. and Narvekar, J., 2005. Seasonal variability of the mixed layer in the central Arabian Sea and its implication on nutrients and primary productivity, Deep-Sea Res pt II., 52(14-15), 1848–1861, doi:10.1016/j.dsr.2005.06.002.
9. Levitus, S., 1982. Climatological atlas of the world ocean. NOAA Prof Pap 13, 1–173.

10. Levitus, S., and T. P. Boyer, World Ocean Atlas 1994, vol. 4, Temperature NOAA Atlas NESDIS J, 117p p., U.S. Govt. Print. Off., Washington, D.C., 1994.
11. Levitus, S., R. Burgett, and T. P. Boyer, World Ocean Atlas 1994, vol. 3, Salinity, NOAA Atlas NESDIS 3, 99 pp., U.S. Govt. Print. Off., Washington, D.C., 1994.
12. Martin S. (2004) An introduction to ocean remote sensing, Cambridge University press, Cambridge, 427 pp.
13. Naqvi, S., Moffett, J.W., Gauns, M., Narvekar, P., Pratihary, A., Naik, H., Shenoy, D., Jayakumar, D., Goepfert, T.J., Patra, P.K., 2010. The Arabian Sea as a high-nutrient, low-chlorophyll region during the late Southwest Monsoon. Biogeosciences.
14. O'Reilly, J.E., Maritorena, S.,Mitchell, B. G., Siegel, D. A., Carder, K. L., Garver, S. A., Kahru, M., & McClain, C. R. (1998). Ocean color chlorophyll algorithms for SeaWiFS, Journal of Geophysical Research 103, 24937-24953, doi: 10.1029/98JC02160.
15. Rao, R.R., Molinari, R.L. and Festa, J.F., 1989. Evolution of the climatological near-surface thermal structure of the tropical Indian Ocean, 1: Description of mean monthly mixed layer depth, sea surface temperature, surface current and surface meteorological fields. Journal of Geophysical Research, 94, 10,801-10,815.
16. Rao, R.R., Sivakumar, R., 2000. Seasonal variability of near-surface thermal structure and heat budget of the mixed layer of the tropical Indian Ocean from a new global ocean temperature climatology. J. Geophys. Res. 105 (C1), 995–1015. http://dx.doi.org/:10.1029/1999JC900220.
17. Ravichandran M., Girishkumar M.S., Stephen Riser, 2012. Observed variability of chlorophyll-a using Argo profiling floats in the southeastern Arabian Sea. Deep-Sea Research I 65 (2012) 15–25.
18. Shalin S., K.V. Sanilkumar (2014) Variability of chlorophyll-a off the southwest coast of India, International Journal of Remote Sensing, 35:14, 5420-5433, DOI: 10.1080/01431161.2014.926411
19. Saha K. (1974). Some aspects of Arabians sea summer monsoon, Tellus, XXVI (4), 464-476.
20. Sarma, Y.V.B., Adnan Al Azri, Sharon L. Smith. 2012. Inter-annual Variability of Chlorophyll-a in the Arabian Sea and its Gulfs. International Journal of Marine Science, Vol. 2, No. 1 doi: 10.5376/ijms.2012.02.0001
21. Singh, A., Gandhi, N., Ramesh, R., 2012. Contribution of atmospheric nitrogen deposition to new production in the nitrogen limited photic zone of the northern Indian Ocean. J. Geophys. Res. Oceans 117.
22. Singh, A., Ramesh, R., 2011. Contribution of riverine dissolved inorganic nitrogen flux to new production in the coastal northern Indian Ocean: An assessment. Int. J. Oceanogr. 2011.
23. Sudheesh, V., Gupta, G., Sudharma, K., Naik, H., Shenoy, D., Sudhakar, M., Naqvi, S., 2016. Upwelling intensity modulates N2O concentrations over the western Indian shelf. J. Geophys. Res. Oceans.
24. Wyrtki, K., 1973. Physical oceanography of the Indian Ocean, in: The Biology of the Indian Ocean. Springer, pp. 18–36.

---

## Author Response (AR1)

Dear Editor,

The authors appreciate the opportunity to improve the manuscript ID bg-2017-285 entitled "Delineation of marine ecosystem zones in the northern Arabian Sea during winter" by Saleem Shalin, Annette Samuelsen, Anton Korosov, Nandini Menon, Björn C. Backeberg and Lasse H. Pettersson. We also thank reviewer for critical reviewing and the comments they provided.

All the concerns of referee are addressed and we hope that the after revisions the manuscript will meet the requirements to be published in the Biogeosciences.

Reply by the authors to the referee's comments is herewith attached.

Looking forward to hear from you,
With regards,

Dr. Shalin Saleem
Research Associate
Central Marine Fisheries Research Institute
Post Box No. 1603, Ernakulam North P.O.,
Kochi-682 018.
Kerala, India

**Reply to Reviewer1:**

On behalf of the authors I thank the reviewer for the critical review of our article. In particular for the suggestion of including chlorophyll images based on climatological data which verifies that our objective classification appropriately categorizes the Arabian Sea ecosystem zones based on Chl-a characteristics for the winter.

Below we respond to each of reviewer's question. Reviewer comments are in green, and our response in black.

**Major comments:**
1. On what basis, principal component analysis based six ecological zones were divided into two Longhurst provinces? It should be elaborated in the section 4.1.
**Authors' comment:** 'Our analysis of 'Chl-a winter variability revealed six distinct ecological zones in the Arabian Sea, which has been compared with the Longhurst biogeographical classification of marine provinces for the study area.' This comparison is done as Longhurst's biogeographic classification is widely accepted for the world oceans including the Indian Ocean. The statement above in quotes will be included in the main text.

2. Because of the lack of satellite data during the monsoon seasons, authors have considered only winter data. We know that the Arabian Sea is most productive during summer. Authors should discuss how ignoring monsoon would impact the delineation of ecological zones?

**Authors' comment:** Yes, this is a limitation of our study, however the amount of comparable high quality (satellite Earth observation) data coverage in time and space make the statistical analysis and zonal classification robust. Combining satellite Earth observation data with seasonal ship measurements would have been advantageous; however we did not have access to such data to be incorporated in our analysis. The most productive period during summer coincides with persistent cloud cover in the study area (Saha, 1974). Since this work utilises satellite ocean colour data, which are limited by cloud cover, ocean colour data cannot be utilised to study the chosen area during the summer (Martin, 2004). Accordingly our analysis focuses on data from the winter period (Nov-March) in order to examine intra- as well as inter-winter Chl-a variability in the study area. With this limitation in our study we clearly state that we have analyzed the intra and inter-winter variability, though the obtained zones would likely be different had it been possible to include the whole year. How it would change is hard to say, but a more prominent signal from the northwest Arabian Sea upwelling region is likely.

3. It appears that only surface chlorophyll values were used. But we know that ocean has a deep chlorophyll maximum (DCM). The justification provided on pages 2-3 (line 34, line 1-2; "the fact that during: : :: : :: : :: : :.content") is not correct. There are numerous studies showing DCM in the Arabian Sea. Authors have covered almost entire Arabian Sea, and it is not possible to have a weak DCM everywhere. Authors should explain how this analysis would be affected by excluding the most productive parts of the ocean?

**Authors' comment:** Yes, the present work utilized depth integrated surface chlorophyll values, which is remotely sensed by ocean colour satellite sensors. Reflected radiance is measured by the ocean colour sensor which contains scattered light containing information from ocean recorded up to the depth where it is no longer reflected back to the surface (i.e., 0.1 photosyntheically available radiation (PAR) depth) (Martin, 2004). Hence remotely sensed Chl-a represents the average of Chl-a concentration from surface up to the depth where 0.1 PAR with that of surface is available.

As pointed out in the manuscript, we admit to the fact that the Arabian Sea does not show weak DCM everywhere during winter (Breves et al. 2003; Revichandran et al. 2012; Prasanakumar 2000). Also, some productivity may be excluded as we are not considering DCM-variability; however, with the absence of in-situ observations, there is no good way to include the deep layers. Furthermore, the increased chlorophyll at depth is sometimes a result of the phytoplankton having higher Chl/C ratio, to compensate for low light, not necessarily higher biomass. Again we will argue that our homogenous data set with extensive coverage in time and space found a basis for robust statistical analysis as long as the limitations of using satellite EO data are taken into consideration.

4. Provide a climatological data based Chlorophyll image as Fig. 4 (c). It would help to see whether chlorophyll content are drastically different in these six zones (particularly for sentence on Page 10, lines 17-18)

**Authors' comment:** Valid comment – Thanks! A Chl-a image (revealing the seasonal average Chl-a values over the winter period (Nov-March) from 2002 to 2013) is provided as a new figure 1, and it can be placed as 4 (c) as per the suggestion. Also, the following sentence provided in quotes is included in the manuscript. 'The annual winter climatology (seasonal average Chl-a values over the winter period (Nov-March) from 2002 to 2013) of Chl-a distribution revealed distinct features for each of the identified ecological zones (Figure R1). Based on the variability of Chl-a concentrations, zone 1 experiences maximum bloom intensity between 1.5 to 9.6 mg m$^{-3}$ with a mean of ~2.6 mg m$^{-3}$ and standard deviation of 0.7 mg.m$^{-3}$. Next to Zone 1, high Chl-a prevails in Zone 2, with a range of 1.4 to 7.0 mg m$^{-3}$ and a mean ~ 2.8 mg m$^{-3}$. Standard deviation observed in Chl-a are same for both zones. Moderate values of Chl-a (1.3 to 1.9 mg m$^{-3}$) are observed in Zone 3, Zone 5 and Zone 6. Though similar range are observed for these three zones, the temporal evolution are different. In zone 3, Chl-a varies between 0.5 to 4.2 mg.m$^{-3}$, with 0.3 mg.m$^{-3}$ standard deviation. Among coastal zones, zone 6 Chl-a standard deviation is high (0.8 mg.m$^{-3}$) with a range of 0.9 to 6.8 mg.m$^{-3}$ than for zone 5 (0.5 mg.m$^{-3}$) between 1.0 to 4.3 mg.m$^{-3}$. Minimum value of Chl-a for the winter is observed in zone 4 (0.2 to 1.2 mg.m$^{-3}$), also in this zone least mean (0.6 mg.m$^{-3}$) and standard deviation (0.2 mg.m$^{-3}$) is observed. The Chl-a geo-spatial statistical variation in the study area clearly demarcates different ecological zones'

[Figure]

Figure R1: Annual winter climatology (seasonal average Chl-a values over the winter period (Nov-March) from 2002 to 2013) of Chl-a revealed from satellite data. The black line indicated the delineated zonal boundaries.

**Minor comments:**
1. Title should be revised as "Delineation of marine ecosystem zones in the northern Arabian Sea during winter"

**Authors' comment:** The suggested title is appropriate for this work and hence it will be changed according to suggestion.

2. Page 3, lines 9-12 can be deleted as they do not provide any info

**Authors' comment:** The sentence is retained, as this sentence connects various supplementary data used in this work.

3. Page 4, lines 12-13, same font should be used for variables

**Authors' comment:** Suggestion will be incorporated in manuscript.

4. Page 4, line 19: (Levitus, 1982) has proposed density criteria to estimate MLD which is used widely (Gardner et al., 1995) and a better criterion than temperature.

**Authors' comment:** We thank the reviewer for this suggestion. The reason we use a MLD based on temperature criteria is because numerous other studies, including Rao et al. 1989; Rao and Sivakumar, 2000 and Kumar and Narvekar, 2005 used MLD based on temperature criteria in the Indian Ocean basin to study the MLD dynamics in the area. By using the same definition of MLD as these authors allows us to compare our results qualitatively to these previous studies. Moreover, a comparison of MLD obtained from the HYCOM modeled data using both temperature as well as density criteria's with the Argo datasets available for winter are carried out. A total of 6256 points are collocated for winter for the entire study area for the comparison. MLD calculated from density criteria have higher RMSD and error percentage (RMSD: 36 m and an error of 68 %) compared with that derived from temperature criteria using 1º C, 0.5 º C and 0.2 º C (RMSD: 20 m and an error of 28 %). This analysis showed better MLD derivation is with temperature criteria. Hence, a second analysis based on different temperature based MLD criteria (1º, 0.5º and 0.2º drop from that at surface) with the Chl-a in the six zones were carried out. From this analysis, it was found that MLD calculated using temperature criteria (1ºC degree) could explain the Chl-a pattern in each of the six selected zones more accurately than those computed using other temperature values. This is the reason for include temperature based MLD in the present work.

5. Page 5, line 18: I should be in italic, in fact all the variables should be made italic throughout

**Authors' comment:** Suggestion will be incorporated in manuscript

6. Page 7, line 4: Oman is an upwelling so how could it be oligotrophic (Wyrtki, 1973)

**Authors' comment:** Thank you for pointing out this error, we replaced the term with "mesotropic". The explanation of the Principal Component Analysis (the corresponding paragraph) is rewritten based on periodicity of Principal Components and hence this sentence will be removed from manuscript.

7. Page 8, line 2: (Naqvi et al., 2010) have not done sampling off Gujarat and Pakistan

**Authors' comment:** Author apologies for the mistake. The reference Naqvi et al. 2010 should be replaced by Sarma et al. 2012. However, this sentence too coincides with Principal Component Analysis which is rewritten and this sentence will be removed.

8. Page 8, line 13: How was coastal Chl a found erroneous?
**Authors' comment:** We used the NASA OBPG chlorophyll-a product derived with the OC4 band ration algorithms performing well only in Case-I waters (see also reply to question 5) (O'Reily et al., 1998). Since the coastal zone is loaded with turbid waters (due to river inflow or resuspension) and may be optically shallow, the OC4 algorithm is not applicable and these zones were excluded from the analysis.

9. Page 9, line 2: blows should be replaced by blow
**Authors' comment:** Suggestion will be incorporated in manuscript

10. Page 9, line 17: "These coastal areas: : :: : :.winter." Reference is needed.
**Authors' comment:** Reference viz. Kumar and Prasad, 1996; Kumar et al. 2001 will be included in the manuscript.

11. Page 10, line 2: zone 6 is also an upwelling region (Sudheesh et al., 2016)
**Authors' comment:** Two reference will be added including Sudheesh et al. 2016 and Shalin and Sanilkumar 2014 in the manuscript

12. Page 10, lines 3-4: "Nutrient supply: : :: : :.zones". Provide reference, perhaps (Singh et al., 2012; Singh and Ramesh, 2011)
**Authors' comment:** The corresponding sentence referee asked is 'Nutrient supply from Narmada and Tapi rivers as well as atmospheric deposition of nitrogen enhances marine production in zone 6.' Authors found nutrient supply from rivers Narmada and Tapi during winter (November to March) is less (Figure R2). The relevant reference on atmospheric deposition on nitrogen in the study area is Singh et al. 2012, which has reported the contribution of atmospheric nitrogen during winter is 0.06 mmol N m-2 day-1 based on 43 in-situ measurements. Assuming that all the six zones are exposed to this level of atmospheric deposition of nitrogen and comparing this with concentration available for the study area (Figure 9 of manuscript), it is clear that the contribution is low. As nitrate contribution is less during winter, this sentence with respect to river discharge and atmospheric nitrate deposition are removed in the revised version.

[Figure]

Figure R2: Annual river discharge from Narmada and Tapi. (Data source: http://nelson.wisc.edu/sage)

**Authors' comment:** Suggestion is included by incorporating the following sentence given in quotes in the manuscript. 'However, we have identified high Chl-a concentration ($>0.5$ mg m$^{-3}$) in the entire study area, with significant differences between various parts, particularly higher values to the waters closer to the coast.'

**Authors' comment:** Suggestion of included lines 5 to 25 from summary to discussion section will be incorporated in the manuscript.

**Authors' comment:** Suggestion will be incorporated in the manuscript in page number 18.

16. It is not clear what Fig. 8 conveys
**Authors' comment:** The sentence provided in quotes will be included in the Figure caption. "Zones 1 to 6 are represented by violet, blue, green, light green, yellow and red lines respectively."

**Authors' comment:** Suggestion will be incorporated in the manuscript.


**Reply to Reviewer2:**

We thank the reviewer for taking the time to provide such detailed comments to our manuscript. Special thanks for the suggestions on $AOT_{ma}$ calculation and the comment on including a figure with wind rose along with DOT. However, we disagree with the reviewer's comment on the lack of merit of the work, particularly the comment that the marine zones can be readily discerned by the spatial distribution of Chl-a. While this is indeed possible, such an approach remains qualitative and subjective, here we provide a more objective and quantitative approach which provides a delineation of the marine ecological zones in the Arabian Sea during winter. Moreover, our method is in agreement with previous literature and we have expanded our analysis to ensure that is robust.

Below we respond to each of reviewer's question. Reviewer comments are in green, and our response in black.

**General comments:** "Can't say I like this paper. The innovative information established by the authors is meager: all prime features of the phytoplankton field across the north Arabian Sea and their driving processes are known and the present research has not contributed to this knowledge. The authors regard as a major merit of their work a more fine delineation of marine zones in the north Arabian Sea as compared to the ones determined previously by other workers. First of all, the zones established by the authors are readily discernible in the spatial distributions of Chl, and secondly, the established contours of the zones are not proven."

**Authors' comments:** In order to ensure the robustness of our delineation of the identified zones a new figure representing the seasonal average of Chl-a over the winter period (Nov-March) is included as Figure R1, which reveals distinct Chl-a characteristics for each of the identified ecological zones. Our objective classification based on winter average of Chl-a values from eleven winter seasons takes into account both spatial and temporal information. To say that the same result could be obtained by the authors by looking at the spatial distribution is highly uncertain and the result would probably depend both on the person doing the subjective analysis and how the data was presented in terms of colormap etc. In the initial manuscript itself, the authors have compared Chl-a variability in six obtained zones with the well-accepted biogeographic classification of Longhurst falling in the selected area. As our study has utilised Chl-a concentration obtained from satellite sensors which has about 100 times finer spatial resolution used by Longhurst for regional mapping for classifying ecological zones in the northern Arabian Sea, this regional classification could delineate the spatial Chl-a variability better with more detailed regional information than obtained from Longhurst's classification. The objectivity of the methods used and the increased amount of information in modern ocean color products are the basis for author's argument about 'finer delineation of marine zones in the north Arabian Sea' is true.

We have analysed physical and chemical characteristics within each of the identified marine ecological zones, and the respective relation between cooling, deepening and production between the six zones. In the analysis section, we have made use of the established, published, knowledge on the driving processes of Chl-a in the study area. Our information is based on surface-data and limited number of variables – hence we must utilize previous studies to better understand our results. However, the in-situ observation coverage in the Arabian Sea is lacking both spatially and temporally and the utilized literature base their result on observations from shorter periods compared to our study. Such long period of information is very essential for resolving inter-annual variability in the ecosystem characteristics. Our study contributes understanding of the temporal/spatial variability of phytoplankton and hence, we disagree with the reviewer's comment on *'The innovative information established by the authors is meager: all prime features of the phytoplankton field across the north Arabian Sea and their driving processes are known and the present research has not contributed to this knowledge'*.

[Figure]

Figure R1: Annual winter climatology (seasonal average Chl-a concentration over the winter period (Nov-March) from 2002 to 2013) of Chl-a revealed from satellite data. The black line indicated the delineated zonal boundaries.

This thesis is underpinned by my comments to the text. The paper composition is also unsatisfactory: instead of partitioning the respective part of the paper into Results and Discussion sections, the authors mixed up the reporting on the results obtained and underpinning of the results' validity. This caused numerous repetitions and unnecessary lengthening of the text. The authors' English needs to be brushed up In light of the above and the comments below, I reckon that the paper should be subsumed under the category "major revision".

We have kept the present format with presenting results and discussion together by topic rather than in separate section, however we have made an effort to clean up the manuscript and avoid repetition. The English has been revised and improved.

We do appreciate the reviewer's suggestion regarding AOT$_{ma}$ calculation and plotting of a wind rose. These two additions in addition to several other comments on poorly written statements have significantly improved our manuscript. However, we disagree with the reviewer's opinion about the PCA and objective analysis. The Chl-a geo-spatial statistical variation in the study area clearly demarcates different ecological zones (see Figure R1 of 4b and additional description related to this figure in the manuscript), which proves the significance of PCA and objective analysis, compared to any subjective method.

**Specific comments**

1. Specify the desert(s); [5 (page 2)]

**Authors' comment:** Arabian desert in the west and Thar desert to the east are the major dust contributing deserts. This sentence is added in Appendix A3.

2. It is insufficient to anticipate: this needs to be proven. [15 (page 2)]

**Authors' comment:** Agreed. The statement has been rephrased and the following references were included in the text that justifies our argument: Longhurst 1995, Longhurst 1998 and Longhurst 2006; Spalding et al. 2012.

3. Why the Chl concentration at 0.5 mg/m3 is used as a criterion? [25 (page 2)]

**Authors' comment:** The concerned statement is a general argument for Chl-a concentration for the study area in an annual cycle (Sarma et al. 2012; Ravichandran et al. 2012). Based on Chl-a monthly climatology for the study area, annual concentration considering all seasons is approximately 0.5 mg m$^{-3}$.

4. Firstly, Mignot et al. reported solely on Pacific and Mediterranean oligotrophic waters (typically, Chl is significantly under 1 mg/m3 ). The actual location and degree of "weakness" of deep Chl maxima (DCM) are site-specific. For the locations within the study waters the assertion that DCM did not affect the satellite-borne Chl concentrations needs independent confirmation. The authors write that DCM in the study area is presumingly shallow because of the strong attenuation by surface Chl. A rather strange argument: if the DCM is shallow then it can be "sensed" by the satellite sensor. Besides, the Chl concentrations reported in your study are not likely to affect the downwelling light to a degree of eliminating the DCM optical influence. At least, a Hydrolight experiment can bring certainty in this issue. [30 (page 2)]

**Authors' comment:** We agree that deep Chl maxima are site-specific. However, some regions in the selected area show shallow DCM (24 m) during winter (Al-Niami et al. 2017), and concurrently regions with deeper DCM exist in the study area (Breves et al. 2003; Ravichandran

et al. 2012; Kumar 2000). Since, it is clear that DCM is not shallow in the entire study area during winter, the statement 'DCM is shallow during winter' is deleted. However, it is to be mentioned here that in-situ coverage on Arabian Sea is not sufficient to give complete spatial and temporal variability on DCM and hence we have to accept the uncertainty on this issue (Barlow et al. 1996).

5. There are no assessments of Chl retrieval errors. This issue is essential, because of the above comment, and also because of the optical heterogeneity within the study waters. It is unnecessary to mention that the NASA algorithm used by the authors is valid (and produces really accurate values of Chl concentrations) only for case I waters (i.e. strictly oligotrophic). However, the authors haven't elucidated this issue with regard to the studied waters in view of the impacts produced by the river discharge, and dust fallouts. The observed variations in Chl could arise, inter alia, from the inability of the NASA algorithm to retrieve Chl correctly in those parts of the study sea where waters are not strictly case I waters. In this case the zoning [in essence, based on Chl variations] might be compromised (at least the declared contours of six zones, which are supposed to be the main advantage of the study). That is why the realistic error bars relevant to the study sea are indispensable for all illustrations of Chl concentrations in the selected zones. The issue of retrieval error arises also with respect to other satellite-borne variables used in the study. [15 (page 3)]

**Authors' comment**: The NASA OBPG chlorophyll product that we used does not have values of uncertainties associated with each value of chlorophyll and, therefore, region-wide assessment of errors in the chlorophyll product is not feasible to perform. The validation shows that in oligotrophic waters the algorithm accuracy is quite high: $r^2 = 0.86$, RMS = 0.25 mg m$^{-3}$ (Feldman, 2017; Hu et al., 2012). Large errors are presumably observed in the turbid waters of the Persian Gulf as well as the coastal areas. Our region of interest excluded coastal areas and included only phytoplankton dominated open ocean areas, where the standard algorithm of NASA is known to work well.

6. As a matter of fact: the coefficients taken from the literature are not necessarily relevant to the study area, e.g. fdu, and AOTm a (the later was determined by Smirnov et al., for Midway Island in the Pacific, located in waters located far away from the study area; meanwhile, it is known that AOTm a depends not only upon the above water surface wind but also on a number of other parameters, that is why there are many parameterizations suggested for specific marine locations). [5 (page 4)]

**Authors' comment:** We thank the reviewer for this comment. This question is a valid one, we were not aware of the stated scenario. It is clear now that in the Indian Ocean an exponential relationship between wind speed and sea salt formation exists, where as in Pacific this relation is

linear. As a result of which we have now replaced Smirnov et al. 2003 with Moorthy 1997 to estimate AOT$_m$. DOT obtained using the new formula is super-imposed in the manuscript figure below (pink line) while red represents DOT as computed with the old formula, as can be seen the values differ, but the temporal evolution is similar in zone 1, 2 and 5 (Figure R2). In the revised manuscript, Figure 9 (figure with DOT) is updated with latest data.

[Figure]

Figure R2: Averaged variability of surface Chl-a, nitrate and DOT in six ecological zones. Viewports (a), (b), (c), (d), (e) and (f) represents variability along first, second, third, fourth, fifth and sixth zones, respectively. Red and pink line indicates DOT computed with the old and new formula.

7. Please, give the major assessments of MLD simulation errors (results of validation by George et al., 2010). Error bars are indispensably required for all illustrations of MLD variations in the selected zones. [20 (page 4)]

**Authors' comment:** Statistical analysis cannot be carried out using George et al. 2010, hence a comparison of MLD modeled data with the recent Argo derived mixed layer climatology (http://mixedlayer.ucsd.edu/Argo) was carried out for winter months. On average a RMSD of 20 m and a 28 % error is observed between model output and Argo dataset. It was found that adding error bars to the plots looked very messy, instead we have added a paragraph in the revised manuscript describing the MLD simulation error in the study region.

8. If only PC 1-3 are meaningful, why you provide illustrations for PC 4 and PC5 (fig. 2). The authors are reporting on the northwestern and southeastern gradients in spatial distributions of PC1 (that is the component that predominantly, accounts for 97% of the spatio-temporal variance in Chl) as one of the important findings. However, this finding could be attained without the PC analysis just by visual examination of the spatial distribution of Chl or/and SST, which is confirmed by the authors themselves. So there is nothing new in this finding. [5(page 7)]

**Authors' comment:** A simple visual examination remains a subjective approach complicated (or, even, disabled) by the fact that sequences of maps of several variables have to be visualized and analyzed simultaneously. The accuracy of such a subjective method is not described anywhere in the literature, nor proven to be correct. In fact, PC1 accounts for only 80% of the Chlorophyll variance.

We developed an objective method of analysis of time series and provide exhaustive explanations of the methodology and description of several experiments to illustrate its sensitivity to various factors (number of PCs, number of zones, etc.).

PC5 is included for illustration of the speckle noise that contaminates the signal. In case of PC4 some of the signal is still present and, therefore, it is used, but PC5 appears to be useless. We respectfully disagree with the comment that there is nothing new in this finding.

9. First, the authors write that PC4 and PC5 are not informative (mostly noise) and then declare that the final delineation into ecological zones was obtained by combining the first 4 PCs. Please, explain. Also, please, explain what you mean saying "based on general Chl pattern in.."[5 and 10 (page 8)]

**Authors' comment:** The write-up on five PC's are rewritten considering the periodicity of these components, which was not done initially. Based on periodicity it is clear that PC4 and PC5 represent the intra-winter variability. Since, this work concentrates on intra-winter variability, these two PC's cannot be considered as noise. The amplitude of PC4 is more than 10% of PC1, and PC4 is necessary to include it in the zoning to get the narrow coastal regions that we know exists and is not present when only 3 PCs are included (see argument in appendix A in the manuscript). These arguments show the reason for including PC4 in the final zoning.

[Figure]

Figure R3: Periodicity for (1) PC4 and (2) PC5.

[Figure]

Figure R4 (Figure 2 from manuscript): Individual maps of principal components (PC 1 to 5) and RGB composite of the first three statistically significant components. Corresponding to each PC, the respectively periodicity is shown

PC4 causes high Chl-a production for January, November months and minima during December (Figure 3R(1)). Most of the high variation during January / November occurs during December too, hence in this PC we can observe more regions with zero variation (indicated by white colour) (Figure 3R(2)). Therefore, the regions under this PC are highly scattered within north, central and eastern part of the study area. Next PC demarcates regions with high Chl-a production for February, and follows a decreasing trend for November to January months. Similar, to PC4, region coming under this PC is also scattered highly. However, this PC differentiates Persian Gulf and Pakistan and Gujarat coast from the rest of the north-central region (Figure R4).

Regarding the second part of the question, a map (RGB composite of the first three statistically significant components) illustrating the significance of combined Principal Components (PCs) is

described (line no. 5). This map is generated with the combination of first three PC's (Figure 2). First PC is represented using red, second by green and third by blue. Zones with similar colors have similar combinations of PC values and therefore this figure illustrates similar winter variability on Chl-a. This image is the application of a statistical clustering method to delineate the study region into areas with distinct Chl-a dynamics. This is based on the values of principal components (details is discussed in section 3 of manuscript).

The same method was applied with the PCs 3, 4 and 5. Clustering in the case, is done making use of the technique 'k-mean Cluster Analysis' (CA). Several combinations of PC and CA is carried out (described in Appendix A). Based on the available knowledge of Chl-a variability as well as oceanographic characteristics in the area the combination of 4 PC and 8 CA is selected (Figure 3 and Appendix A in the main manuscript).

10. Please, explain, on the basis of what it was decided that satellite-derived Chl values along coastal and shallow waters were erroneous.

**Authors' comment:** Retrieval of Chl a concentration from optical satellite data near to the coast is complex. The water masses contain optical properties of riverine fresh water influx, containing terrogeneous dissolved organic compounds, and these contribute to an error in Chl-a retrieval. In addition, shallow water depth regions (depth < 30 m) may include signals from bottom reflections (with clear waters conditions), which can introduce additional errors in the retrieval process (Martin, 2003). Considering the fact that our present work uses a global Chl-a retrieval algorithm (OCI) to obtain Chl-a along with the above mentioned two points, we mask out the analysis of satellite derived Chl-a values near the coast. Additionally, in our response to question number 5 we explain that only regions classified as case-1 waters during winter in the selected study area, where the NASA algorithm will work well, are selected.

11. Please, explain in the paper what are the reasons to believe that " the physical forcing affecting chl concentration along the two regions is likely to be different" … [10 (page 8) 15 (page 8)]

**Authors' comment:** Based on knowledge available through published studies (Kumar and Prasad, 1994; Kumar et al. 2000; Shetye et al. 1994) it is concluded that the two regions are likely to be different. Accordingly, these references will be included in the revised manuscript.

12. The authors write that 1-3 zones (encompassed by Longhurst's ARAB zone) are strong upwelling regions with high Chl in winter time, and then they refer to Longhust who defines the ARAB province as a zone with strong upwelling during summer and strong convective cooling during winter. Obviously, some phrase is required to follow these statements in order to clarify the actual hydrodynamic situation therein. [5 and 10 (page 9)]

**Authors' comment:** We thank the reviewer for pointing out this mistake, and have corrected the text accordingly. Zones 1-3 are regions where strong convective overturning occurs during winter (page nos 9 (line number 8, 12 and 19), 10 (line number 1) and 11 (line number 20).

Hence, the comparison of Chl in the convective zones identified with Longhurst province during winter has been carried out.

13. Please, specify 1. what is known about the atmospheric deposition on nitrogen (there is no respective reference), and 2. why this mechanism of nutrient supply acts only in zone 6 (or, at least, is not mentioned with regard to other zones). Also, specify the annual cycle of stream flow of the Narmada and Tapi rivers to support your thesis that nutrient supply from Narmada and Tapi rivers as well as atmospheric deposition of nitrogen enhances marine production in zone 6. This additional information might clarify the authors' statement that in zone 6 "peak Chl-a is observed during January" as opposed to other zones.[ 5 (page 10)]

**Authors' comment:** A relevant reference on atmospheric deposition on nitrogen in the study area is Singh et al. 2012. They have reported the contribution of atmospheric nitrogen during winter is 0.06 mmol N m-2 day-1 based on 43 in-situ measurements. Assuming that all the six zones are exposed to this level of atmospheric deposition of nitrogen and comparing this with concentration available for the study area (Figure 9 of manuscript), it is clear that the contribution is low. The complexity of zone 6, in particular, can be explained by additional sources of nitrogen supply e.g., likely from rivers discharges (see Figure R5). The sentence will be restructured accordingly.

[Figure]

Figure R5: Annual river discharge from Narmada and Tapi. (Data source: http://nelson.wisc.edu/sage)

14. First, the authors write that the inverse relationship between SST and Chl-a have weak correlation coefficient 1 in zone 1 (r = 0.39, n=60) and zone 2 (r = 0.55, n=60). Then a bit further: "However, MLD and Chl-a in zone 1 and 2 are moderately correlated (correlation coefficient, r = 0.28)". What are your criteria in this regard? [15 (page 11)]

**Authors' comment:** We thank the reviewer for this observation, as you pointed out a criteria based on r value is defined as follows:

- r> 0.50 is high,
- r>0.35 is moderate
- r< 0.35 is low.

Which will be introduced in the manuscript. Hence, the above mentioned statement will changed as 'the inverse relationship between SST and Chl-a have moderate correlation coefficient 1 in zone 1 (r = 0.39, n=60) and zone 2 (r = 0.55, n=60).' "However, MLD and Chl-a in zone 1 and 2 are poorly correlated (correlation coefficient, r = 0.28)".

15. The authors write "Mean wind speed in zone 1 is highest during January (3 m s−1) and in zone 2 during December (> 3 m s−1) (Figure 5a"). Does fig. 5a collaborates this statement? Further on: "During November to December, low PAR (33-36 E m−2 day−1) prevailed in the study area, corresponding to low temperature and enhanced mixing, deepening the MLD. But according to fig. 5 in November –December MLD is still rather shallow, especially in November. [25 (page 11)]

**Authors' comment:** Thanks for the critical observation. The sentence is rephrased now as 'Wind speed fluctuates strongly for zones 1 and 2. In zone 1, maximum variability (0.5-3.0 m s$^{-1}$) is seen during November and December and for zone 2, wind varies strongly throughout winter, with maximum wind speed (0.5-3.0 m s$^{-1}$) for December and January months.' This sentence is corroborated by the data presented in Figure 5A.

Regarding the second suggestion, the sentence is changed to 'Decreasing pattern in PAR (33-36 E m$^{-2}$ day$^{-1}$) prevailed in the study area during November to December for both zones, which corresponds to a reducing trend in temperature and deepening MLD cycle.'

16. The fig. captions are poorly written: "Time series of the monthly average concentration of wind speed and PAR (a1 and a2) SST and MLD" [5 (page 12) and 5 (page 13)]

**Authors' comment:** The figure caption has been rephrased:
"Temporal variability of wind speed and PAR (a), SST and MLD (b) and surface Chl-a (c) averaged for zone 1 (left, denoted by suffix 1) and zone 2 (right, denoted by suffix 2) during the winter period for the years 2002–2013. Pink colour is used to represent Chl-a, SST and wind speed and blue to represent MLD and PAR. Thick lines represents mean and the shaded areas the standard deviation for each parameter. The time series for the individual years are shown using thin lines. Vertical dotted lines represent the timing (month) of peak algae blooms in each zone."

17. Please, comment on your finding that PAR and Chl for zone 5 are not correlated at all, and for zone 6 they are inversely correlated. Also, some interpretational comments are required for

the phrase "For zone 5, wind and Chl-a production are weakly correlated (r =0.30, n=60), while in zone 6, these parameters are not correlated (r = -0.09, n=60)" [5 (page 16)]

**Authors' comment:** MLD in Zone 5 are ~30 - 40 m shallower than in zone 6 and hence strong winds for the entire month will have triggered mixing, supplying more nutrients than by convective mixing alone to the mixed layer enhancing Chl-a production. In zone 6, wind fluctuates strongly compared to zone 5. Zone 6 is classified as INDW in Longhurst's classification, where wind induced blooms are observed. However, the time scale of wind induced bloom, will be of the order of days / weeks and not months and hence on monthly scales, the wind's influence will not be resolved.

18. why the regression equations do not include such variables as MLD, concentration of nitrates nitrates and iron. It would be much better to do so instead of discussing the relations between Chl and the above variables apart from the variables reflected in Table 1. [Table1]

**Authors' comment:** Linear regression as well as the multiple regression analysis is done utilizing monthly data. Whereas, the nitrates and iron data is available only as monthly climatology. Therefore, regression analysis on monthly scale cannot include nitrate or iron concentration using the available data.

19. Caption for Fig. 8 lacks the designations of colours. [Page 17]

**Authors' comment:** Sentence provided in quotes will be included in the Figure caption. "Zones 1 to 6 are represented by violet, blue, green, light green, yellow and red lines respectively."

20. Please, give (at least in the Appendix section) the rose of winds in winter in order to let the reader better understand why in some parts of the sea DOT is higher than in the others. It would be good to give alongside it the field of DOT over the study area. [15 (page 13)]

**Authors' comment:** As suggested we have plotted the wind roses for the respective zones in order to reveal the possible source locations of DOT (Figure R6). For Zone 1, both the Thar desert and Arabian desert contribute to DOT, as the strong winds have directions between northerly to north-westerly. Similarly for zone 2, both these zones can be significant. While, for zone 3, its the Arabian desert contribution more to DOT enhancement as revealed from wind rose diagram. While for zone 4, it's the continental wind from Indian sub-continent available in the area. This is consistent with Patel et al. 2017.

[Figure]

Figure R6. Wind rose diagram for the six zones. Zone number corresponding to wind rose plot is provided.

21. As was commented above, the reported finding on the north-south gradient in Chl is stale and had been established without any complicated processing procedures. The same comment can be made with regard to the identified number of [10 (page 19)]

**Authors' comment:** The north-south gradient in Chl-a is visible in satellite images (individual and binned data, however identification of other PC's contributing to Chl-a distribution during winter cannot be done with subjective / visual analysis. An objective method is required to handle it and in this paper we have elaborated a method using the combination of principal component analysis and cluster analysis. The number of differentiated zones in the region is consistent with what is found in literature for other marine areas.

22. The reported finding that "The increased amount of Chl-a production in the open ocean zones are found to be directly related to sea surface temperature variability (ie. cooling) and the deepening of the mixed layer " is neither an unknown phenomenon for the study area. [5 (page 20)]

**Authors' comment:** The sentence has been modified: "In agreement with other studies, an increase in the concentration of Chl-a in the open ocean zones (zones 1, 2, 3 and 4) are found to

be directly related to the variability of the sea surface temperature (ie. surface cooling) and the deepening of the mixed layer.

23. "The combined analysis of DOT and nitrate suggests that the variability of the algae blooms depend on both sources in these zones. The variability of Chl-a in the northern and northwestern parts of the Arabian Sea is correlated strongly with the atmospheric deposition of iron from January to March" The two statements appear kind contradictory. [15(page 20)]

**Authors' comment:** The initial sentence is in general for all zones, a dependence of both parameters is observed in six zones. However, in the north and northwestern Chl-a production and DOT follows similar trend of variation and hence in these zones [zone 1, 2, 3 and 5] strongly is observed with the atmospheric deposition of iron from January to March.

**Clarification of the sentence, according to comment**; "The combined analysis of DOT and nitrate suggests that the variability of the algae concentration depend on both sources of nutrient supply in all six identified ecological zones. However, the variability of Chl-a in the northern and northwestern parts of the Arabian Sea (zones 1, 2, 3 and 5) is predominantly correlated with the atmospheric deposition of iron during the period from January to March."

24. It is difficult to agree with the authors' statements that "This study provides a more comprehensive understanding of the environmental factors controlling the spatio-temporal variability of the marine chlorophyll a concentration in the northern Arabian Sea during winter conditions", and further on "Additionally, this study reveals the need for better understanding of factors controlling the marine primary productivity in other coastal upwelling zones". Indeed, to justify/prove the validity of each zone the authors refer to the relevant publications of other workers who investigated in depth the factors and mechanisms controlling the spatiotemporal variability of the marine chlorophyll a concentration. Also, in many studies of the north Arabian Sea the need of further investigations, and more thorough sampling/in situ determinations of physico- and biogeochemical variables. [30 (page 20)]

**Authors' comment:** Distinct Chl-a characteristics for each of the identified ecological zones clearly indicates the spatial variation of Chl-a during winter is better brought out in this work (Figure R1). The temporal variability in Chl-a in the six delineated zones is an objective way to study spatio-temporal variability. This comparison clearly indicated the significance of the present classification.

Authors have correlated the surface cooling and mixed layer deepening with Chl-a production, in the delineated six ecological zones, which is required to explain Chl-a characteristics in these zones. Distinct physical and chemical characteristics within zones are identified. For the first five zones, it's the cooling followed by MLD deepening which enhances nutrient availability resulting in increased production, in zone 6 while MLD deepening / Chl-a production is followed by maximum cooling. In the analysis section, we have used the established knowledge on driving processes of Chl-a processes in the study area based on published information. Such comparisons

are entirely appropriate for research and development. Also, the complex influence of both nutrient and DOT between north, north-east as well as south and southeast part of the study area is brought out in this work. Though, similar studies have been carried out, these have been done for smaller areas and shorter periods, while our study has covered eleven years of data covering the entire Arabian Sea. We have analysed the influence of DOT and nitrate in all zones and found that in the north and northeast production is strongly influenced by high DOT, rather than nitrate availability. Therefore, we respectfully disagree with the reviewer's opinion that this study does not provide a more comprehensive understanding of the environmental factors controlling the spatio-temporal variability of the marine chlorophyll a concentration in the northern Arabian Sea during winter conditions. We show that it does.

Furthermore, the in-situ coverage in the Arabian Sea is not great, hence we argue that the spatial and seasonal distribution of physical mechanism coupled with production in the Arabian Sea is fully known yet.

[revised manuscript text omitted]

---

## Author Response (AR2)

Dear Editor,

The authors are thankful to the Editor for providing the opportunity to improve the manuscript ID bg-2017-285 entitled "Delineation of marine ecosystem zones in the northern Arabian Sea using an objective method" by Saleem Shalin, Annette Samuelsen, Anton Korosov, Nandini Menon, Björn C. Backeberg and Lasse H. Pettersson. We also thank the reviewers for the comments they provided.

All the concerns of referees have been addressed and we hope that the revised manuscript will meet the requirements to be published in the Biogeosciences.

Reply by the authors to the referee's comments is given below.

Looking forward to hear from you,
With regards,

Dr. Shalin Saleem
Research Associate
Central Marine Fisheries Research Institute
Post Box No. 1603, Ernakulam North P.O.,
Kochi-682 018.
Kerala, India

**Reviewer 1:**

Authors should explicitly admit that they are not able to assess the accuracy of location of the delineating lines they proposed.

**Reply to Reviewer1:**

On behalf of the authors I thank the reviewer for the critical review the article. The article is being modified as per the suggestion. We write about the limitations of assessing accuracy of location of the delineated lines due to the scarcity of *in-situ* observations. Hence, we have included sentences stating this at pages 9 (line number: 2 – 4), 11 (line number: 28 – 30) and 21 (line numbers: 8 and 32).

**Reviewer2:**
References and language should be checked before final decision.

**Reply to Reviewer2:**

On behalf of the authors I thank the reviewer for critically reviewing the article. The paper is thoroughly checked for grammatical mistake by all authors. The changes made are provided in red font. The reference formats have also been cross-checked, two references were left out and they are now added. Also, the references are crosschecked with the format provided through                                                                                                 website (https://www.biogeosciences.net/Copernicus_Publications_Reference_Types.pdf).

[revised manuscript text omitted]